# BCAS2 is involved in alternative mRNA splicing in spermatogonia and the transition to meiosis

Wenbo Liu[1,2], Fengchao Wang[3], Qianhua Xu[2], Junchao Shi[2], Xiaoxin Zhang[2], Xukun Lu[2], Zhen-Ao Zhao[2], Zheng Gao[2], Huaixiao Ma[2], Enkui Duan[2,4], Fei Gao[2,4], Shaorong Gao[5], Zhaohong Yi[6] & Lei Li[2,4]

Breast cancer amplified sequence 2 (BCAS2) is involved in multiple biological processes, including pre-mRNA splicing. However, the physiological roles of BCAS2 are still largely unclear. Here we report that BCAS2 is specifically enriched in spermatogonia of mouse testes. Conditional disruption of *Bcas2* in male germ cells impairs spermatogenesis and leads to male mouse infertility. Although the spermatogonia appear grossly normal, spermatocytes in meiosis prophase I and meiosis events (recombination and synapsis) are rarely observed in the BCAS2-depleted testis. In BCAS2 null testis, 245 genes are altered in alternative splicing forms; at least three spermatogenesis-related genes (*Dazl*, *Ehmt2* and *Hmga1*) can be verified. In addition, disruption of *Bcas2* results in a significant decrease of the full-length form and an increase of the short form (lacking exon 8) of DAZL protein. Altogether, our results suggest that BCAS2 regulates alternative splicing in spermatogonia and the transition to meiosis initiation, and male fertility.

[1] School of Life Sciences, University of Science and Technology of China, Hefei 230026, China. [2] State Key Laboratory of Stem Cell and Reproductive Biology, Institute of Zoology, Chinese Academy of Sciences, Beijing 100101, China. [3] National Institute of Biological Sciences, Beijing 102206, China. [4] University of Chinese Academy of Sciences, Beijing 100049, China. [5] School of Life Sciences and Technology, Tongji University, Shanghai 200092, China. [6] Key Laboratory of Urban Agriculture (North) of Ministry of Agriculture, College of Biological Science and Engineering, Beijing University of Agriculture, Beijing 102206, China. Correspondence and requests for materials should be addressed to S.G. (email: gaoshaorong@tongji.edu.cn) or to Z.Y. (email: yizhaoh2009@163.com) or to L.L. (email: lil@ioz.ac.cn).

Alternative pre-mRNA splicing is critical for post-transcriptional regulation of gene expression, during which particular exons from the same pre-mRNA might be excluded, included or modified to produce multiple mature mRNAs, often in an organ-, tissue- or cell-type-specific manner[1–3]. Thus, alternative splicing significantly expands the form and function of the genome of organisms with limited gene number and is especially important for highly complex organisms and tissues[4,5]. Highly complex tissues, such as the testis and brain, have more gene splicing variants than any other tissues[4,6,7]. In mouse testis, spermatogenesis is a complex process involving mitotic cell division, meiosis and spermiogenesis to give rise to haploid spermatozoa. Alternative splicing variants, especially exon-skipping forms, are enriched in several stages of mouse spermatogenesis[7–9]. In addition, a number of trans-acting regulators of pre-mRNA splicing are primarily or exclusively expressed in the testis[9,10].

Substantial evidence suggests that pre-mRNA splicing is an important regulator of mouse spermatogenesis. Although the roles of most spliced forms of specific genes during this process are unclear, several genes important for spermatogenesis have specific splice variants in different developmental stages. For example, *c-kit* is specifically expressed in differentiating spermatogonia and is essential for the survival and proliferation of pre-meiotic germ cells[11–13]. However, the truncated form of *c-kit* (*tr-kit*) is expressed in the post-meiotic stages of spermatogenesis and enriched in spermatozoa, and could trigger parthenogenesis activation when injected into the cytoplasm of MII oocytes[14,15]. In addition, numerous trans-acting regulators of pre-mRNA splicing are important for spermatogenesis. For example, *Ptbp2*, a key alternative-splicing regulator in the nervous system, is critical for male germ cell survival and male fertility through regulating the proper alternative splicing of germ cell messenger RNAs (mRNA) in the testis[16]. *Rbm5*, a male germ cell splicing factor, is essential for the appropriate alternative splicing of pre-mRNAs involved in spermatid differentiation[17]. *Ranbp9* (Ran-binding protein 9) is also involved in regulating the proper splicing pattern of some spermatogenic mRNAs by interacting with several essential splicing factors (e.g., SF3B3 and HNRNPM) and poly (A) binding proteins (PABPs)[18]. Despite protracted effort, deciphering how alternative pre-mRNA splicing functions during spermatogenesis remains a great challenge for the field.

Breast carcinoma amplified sequence 2 (BCAS2) is preferentially known as pre-mRNA splicing factor SPF27 and was originally characterized as an up-regulated gene by amplification in human breast cancer cells[19,20]. Subsequent studies reveal that BCAS2 is a core component of the CDC5L/Prp19 complex[21]. The Prp19 complex is highly conserved and is involved in the assembly and conformation of the spliceosome, especially important for the catalytic activation of the spliceosome[21–23]. Mutation of the yeast BCAS2 ortholog Cwf7 or Snt309 results in the accumulation of pre-mRNA[24,25]. In *Drosophila melanogaster*, BCAS2 is essential for viability and may function in pre-mRNA splicing[26,27]. BCAS2 has also been shown to be involved in DNA damage repair through the replication protein A (RPA) complex in various cell lines[28,29]. Recently, we found that maternal BCAS2 responds to endogenous and exogenous DNA damage in mouse zygotes and maintains the genomic integrity of mouse early embryos through RPA (ref. 30). Thus, the roles of BCAS2 are context-specific depending on the model system. Currently, the physiological function of BCAS2 is still largely unclear.

In this study, we found that BCAS2 was comparatively enriched in spermatogonia of the mouse testis. Disruption of BCAS2 in germ cells with *Vasa-Cre* led to male infertility, but has little effect on spermatogonia. Although the spermatogonia were grossly normal, spermatocytes in meiosis prophase I were scarce

and meiosis events did not occur in the BCAS2-depleted testis. We further showed that BCAS2 was involved in pre-mRNA splicing in spermatogonia in the mouse testis. Our data reveal a critical role of BCAS2 involving in pre-mRNA splicing of spermatogonia and the transition to meiosis, and male fertility.

## Results

**The expression of BCAS2 in mouse testes.** To explore the potential function of BCAS2 in mouse spermatogenesis, we first examined the expression of BCAS2 in the testis by immunostaining with rabbit anti-BCAS2 antibody. BCAS2 was expressed in the nucleus of both germ cells and somatic cells during testis development (Fig. 1a). Interestingly, in embryonic day 15.5 (E15.5) and newborn mouse testes, BCAS2 expression was relatively high in the prospermatogonia located in the centre of the seminiferous tubules of the testes. At postnatal day 5 and 14 (P5 and P14), BCAS2 was enriched in certain cells located in the basement membrane (Fig. 1a).

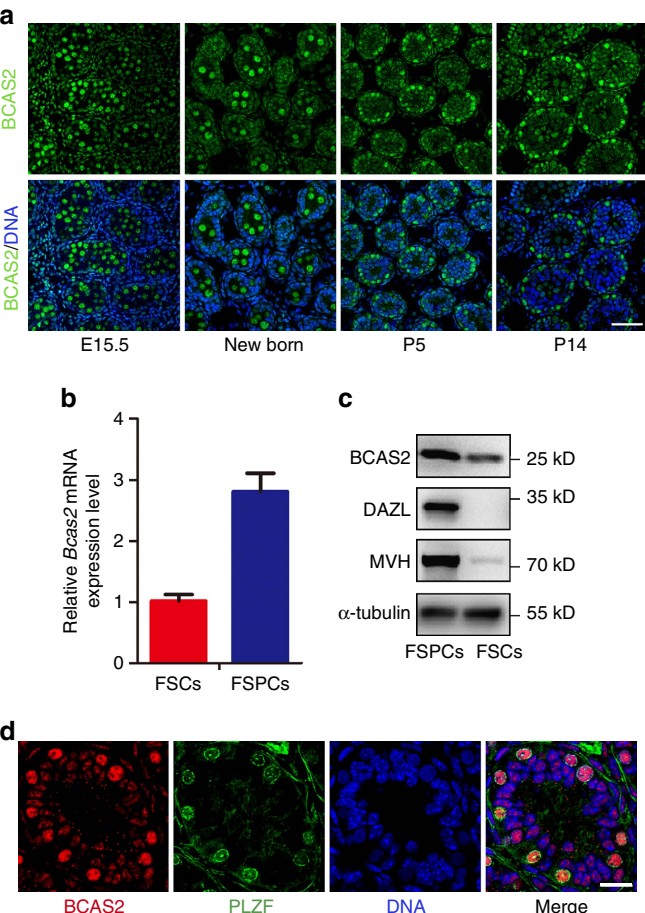

**Figure 1 | Expression of BCAS2 in male mouse germ cells.**
(**a**) Immunofluorescence (IF) staining of BCAS2 in the paraffin sections of testes from E15.5 to P14 mice. The DNA was stained with Hoechst 33342. Scale bar, 50 µm. (**b**) Real-time PCR analysis of *Bcas2* expression in the fraction of spermatogenic cells (FSPCs) and the fraction of somatic cells (FSCs) enriched from P9 testes. *Gapdh* was used as the internal control for normalization ($n = 4$). Error bars represent s.e.m. (**c**) Western blotting analysis of BCAS2 expression in the fraction of spermatogenic cells (FSPCs) and the fraction of somatic cells (FSCs) enriched from P9 testes. Germ cell markers (DAZL and MVH) were used as the indicator of the enrichment efficiency and α-tubulin was used as the loading control. (**d**) Paraffin sections of P8 testes were co-stained with rabbit anti-BCAS2 and mouse anti-PLZF antibodies. The DNA was stained with Hoechst 33342. Scale bar, 20 µm.

Then, we used a differential adhesion method to enrich spermatogenic cells and somatic cells from the testes of P9 mice based on that somatic cells are mainly attached to the bottom of the culture plate, while spermatogenic cells are enriched in the suspended cells[31]. Real-time RT-PCR and flow cytometry analysis using germ cell and somatic cell markers showed that the enrichment was efficient (Supplementary Fig. 1a,b). Next, we examined the expression of *Bcas2* in these two types of cells using real-time RT-PCR and Western blotting. Our results showed that both mRNA and protein levels of *Bcas2* were relatively high in the fraction of spermatogenic cells (FSPCs) (Fig. 1b,c). To further investigate the identity of the BCAS2-enriched cells, we co-stained BCAS2 with the spermatogonia-specific transcription factor PLZF (Promyelocytic leukaemia zinc finger) in P8 mouse testes. The expression of BCAS2 was comparatively enriched in PLZF-positive cells (Fig. 1d).

**BCAS2 is essential for male fertility and spermatogenesis.** To investigate the function of *Bcas2* in spermatogenesis, *Bcas2* was specifically deleted from mouse germ cells by crossing *Bcas2$^{Floxed/Floxed}$* (*Bcas2$^{F/F}$*) mice[30] with *Vasa-Cre* transgenic mice in which the recombinase is specifically active in germ cells as early as E15.5 (ref. 32). The results of real-time RT-PCR (Fig. 2a) and Western blotting (Fig. 2b,c) showed that BCAS2 was reduced in P8 testes of *Bcas2$^{F/-}$;Vasa-Cre* mice. BCAS2 staining showed that the protein was dramatically decreased in PLZF-positive cells from *Bcas2$^{F/-}$;Vasa-Cre* testes (Fig. 2d). Thus, we successfully established male germ cell-specific knockout mice for *Bcas2*, as early as P8 in the spermatogonia.

*Bcas2$^{F/-}$;Vasa-Cre* male mice developed to form grossly normal adults. Although copulatory plugs were routinely observed, no pups were obtained when adult *Bcas2$^{F/-}$; Vasa-Cre* males mated with normal fertile females (Table 1). Compared with controls, the testes of adult (more than two-month-old) and one-month-old *Bcas2$^{F/-}$;Vasa-Cre* males were much smaller (Fig. 2e; Supplementary Fig. 2a) and the testis weight was significantly lower (Fig. 2f; Supplementary Fig. 2b). We next analysed the histology of the testes from the adult and one-month-old males by hematoxylin and eosin (H&E) staining. The seminiferous tubules in the control testes contained a basal population of spermatogonia, several types of spermatocytes and spermatids. However, germ cells were severely reduced in number, and no spermatocytes and spermatids were observed in the testes of *Bcas2$^{F/-}$;Vasa-Cre* males (Fig. 2g; Supplementary Fig. 2c), indicating impaired spermatogenesis in these males. Immunofluorescence results revealed that the number of germ cells marked by MVH was dramatically lower in *Bcas2$^{F/-}$; Vasa-Cre* testes, with only a few MVH-positive cells around the basement membrane in the testes of adult and one-month-old mice (Fig. 2h and Supplementary Fig. 2d). Taken together, these data demonstrate that BCAS2 plays a critical role during normal mouse spermatogenesis and is essential for male fertility.

**BCAS2 is required for the initiation of meiosis in males.** To investigate spermatogenesis in *Bcas2$^{F/-}$;Vasa-Cre* males, we examined the expression of MVH in testes during early development at P10, P12 and P15. Western blotting showed that the expression of MVH in *Bcas2$^{F/-}$;Vasa-Cre* testes of P10 mice was comparable to the control, but was lower in P12 testes and dramatically reduced in P15 testes from *Bcas2$^{F/-}$;Vasa-Cre* mice (Fig. 3a). Consistently, immunofluorescence staining showed that MVH-positive cells in the centre of the seminiferous tubules were apparently lost as early as P12, but MVH-positive cells around the basement membrane were present until P15 in *Bcas2$^{F/-}$; Vasa-Cre* testes, suggesting that meiosis was impaired in

*Bcas2$^{F/-}$;Vasa-Cre* testes (Fig. 3b). To further investigate the meiosis deficiency, we analysed meiotic recombination and synapsis formation in *Bcas2$^{F/-}$;Vasa-Cre* testes with event-specific markers, including γH2AX and SCP3 (ref. 33). Western blotting result showed that the expression of SCP3 and γH2AX was dramatically decreased in *Bcas2$^{F/-}$;Vasa-Cre* testes at P10 and P12 (Fig. 3c). Immunofluorescence staining result confirmed that very few or no SCP3/γH2AX-positive cells were observed in the seminiferous tubules of *Bcas2$^{F/-}$;Vasa-Cre* testes at P12 (Supplementary Fig. 3a,b). These data further suggest that *Bcas2* is required for meiosis prophase I in mouse spermatogenic cells.

Germ cells progress through leptotene, zygotene, pachytene and diplotene during the development of primary spermatocytes. This period is easily distinguished by H&E staining according to the DNA state[34,35]. To further investigate the defects in meiosis, *Bcas2$^{F/-}$;Vasa-Cre* and control testes at similar ages between P10 and P15 were fixed and stained with H&E. In control testes, germ cells in some seminiferous tubules initiated meiosis and started to progress into leptotene or early pachytene at P10 (Fig. 3d), many spermatocytes entered into zygotene at P12 (Fig. 3e) and pachytene at P15 (Fig. 3f). However, leptotene, zygotene and pachytene spermatocytes were almost absent in the seminiferous tubules of *Bcas2$^{F/-}$;Vasa-Cre* testes from P10 to P15 (Fig. 3g–i). In P10 *Bcas2$^{F/-}$;Vasa-Cre* testes, only a few germ cells were in preleptotene (Fig. 3g), which is the last microscopically defined period before meiotic prophase[34]. On the other hand, we also observed few cells with typical pyknotic nuclei in the centre of the seminiferous tubules in *Bcas2$^{F/-}$;Vasa-Cre* testes at P12 and P15 (Fig. 3h,i; red arrows), most likely indicating that those few germ cells might be arrested in early prophase I. Taken together, these data indicate that germ cells without functional BCAS2 may fail to enter meiotic prophase.

Next, we examined the expression of marker genes that were specific and essential for the development of spermatogonia and meiosis prophase I at P9, when the histological experiments could not distinguish *Bcas2$^{F/-}$;Vasa-Cre* testes from the controls. The mRNA expression level of genes involved in spermatogonia proliferation and/or differentiation (*Pou5f1*, *Sox2*, *Nanos2*, *Mvh*, *Sohlh1*, *Sohlh2*, *Ngn3*, *c-kit*, *Dazl* and *Stra8*) was similar in both control and *Bcas2$^{F/-}$;Vasa-Cre* testes. However, compared with the control, the mRNA expression of genes essential for meiosis prophase I (*Dmc1*, *Scp1*, *Scp3*, and *Smc1b*) was significantly lower in the *Bcas2$^{F/-}$;Vasa-Cre* testes (Fig. 3j). Altogether, these data suggest that BCAS2 is required for the initiation of meiosis prophase I in male mouse germ cells.

**Spermatogonia appear grossly normal in *Bcas2* null males.** The MVH positive cells in the basement membrane persisted in the P15 *Bcas2$^{F/-}$;Vasa-Cre* testes, suggesting that the spermatogonia might be normally persistent in these mice (Fig. 3b). To reduce the possibility of an indirect defect of BCAS2 loss on spermatogonia, we stained PLZF with a specific antibody in sections from early developmental testes at P12. The distribution of PLZF-positive cells was similar in *Bcas2$^{F/-}$;Vasa-Cre* testes and the control (Fig. 4a). Furthermore, the average number of PLZF-positive cells per seminiferous tubule cross-section in the *Bcas2$^{F/-}$;Vasa-Cre* testes was not significantly different from the control testes (Fig. 4b). These data suggest that the location and number of spermatogonia are not obviously affected in *Bcas2$^{F/-}$;Vasa-Cre* testes.

Next, we investigated the proliferation and apoptosis of spermatogonia in *Bcas2$^{F/-}$;Vasa-Cre* testes. To examine the mitotic status of spermatogonia in *Bcas2$^{F/-}$;Vasa-Cre* testes at P12, we simultaneously stained the testes for PLZF and Ki67. The number of both Ki67$^+$ PLZF$^+$ cells in *Bcas2$^{F/-}$;Vasa-Cre* testes was not significantly different from the control testes

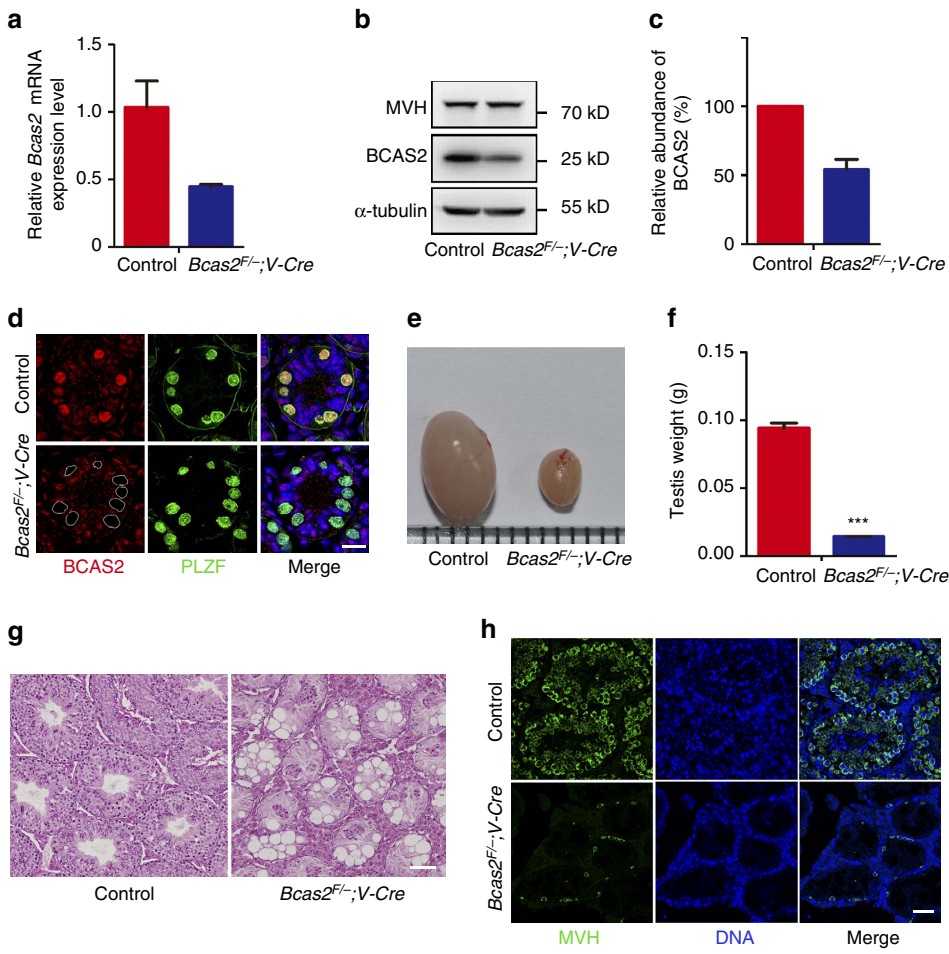

**Figure 2 | BCAS2 is required for germ cell development and male fertility.** (**a**) Real-time RT-PCR analysis of *Bcas2* mRNA levels in control and *Bcas2^{F/−}*;*Vasa-Cre* (*Bcas2^{F/−}*;*V-cre*) testes of P8 mice with *Hprt* as the internal control (*n* = 3). Error bars represent s.e.m. (**b**) Western blotting analysis of BCAS2 protein in control and *Bcas2^{F/−}*;*Vasa-Cre* testes of P8 mice. MVH and α-tubulin was used as a germ cell marker and loading control, respectively. (**c**) Relative abundances of BCAS2 in the control and *Bcas2^{F/−}*;*Vasa-Cre* testes of P8 mice were determined by Western blotting analyses from 4 independent experiments. Error bars represent s.e.m. (**d**) IF staining of BCAS2 in the control and *Bcas2^{F/−}*;*Vasa-Cre* testes of P8 mice. White circles denote the BCAS2 null spermatogonia. PLZF was co-stained to indicate the location of spermatogonia. The DNA was stained with Hoechst 33,342. Scale bar, 20 μm. (**e**) Morphological analysis of adult testes showed that the *Bcas2^{F/−}*;*Vasa-Cre* testes were smaller than the control. (**f**) Testes weight of adult control and *Bcas2^{F/−}*;*Vasa-Cre* mice (***P < 0.001, *n* = 5). Error bars represent s.e.m. (**g**) Hematoxylin and eosin (H&E) staining of adult testes in control and *Bcas2^{F/−}*;*Vasa-Cre* mice. Spermatocytes and spermatids were almost absent in the seminiferous tubules of the *Bcas2^{F/−}*;*Vasa-Cre* mice. Scale bar, 100 μm. (**h**) IF staining of MVH (a germ cell marker) in adult testes in control and *Bcas2^{F/−}*;*Vasa-Cre* mice. Compared with the control, only a few MVH positive cells in the basement membrane were observed in *Bcas2^{F/−}*;*Vasa-Cre* testes. The DNA was stained with Hoechst 33342. Scale bar, 50 μm.

| Table 1 | The fertility of *Bcas2^{F/−}*;*Vasa-Cre* males. | | | | |
|---|---|---|---|---|---|
| **Genotype** | | **No. of male mice** | **No. of plugged female mice** | **No. of litters** | **No. of pups per litter** |
| **Male** | **Female** | | | | |
| Control | Wt | 5 | 24 | 23 | 11.92 ± 0.5 |
| *Bcas2^{F/−}*;*V-Cre* | Wt | 5 | 23 | 0 | 0 |

(Fig. 4c,d), suggesting that the proliferation of spermatogonia was normal in *Bcas2^{F/−}*;*Vasa-Cre* testes. To analyse the apoptosis of spermatogonia, we performed double staining of PLZF and cleaved caspase 3 (CAP3) in *Bcas2^{F/−}*;*Vasa-Cre* testes. Compared with the control, the number of CAP3^+ PLZF^+ spermatogonia was not significantly higher in *Bcas2^{F/−}*;*Vasa-Cre* testes (Fig. 4e,f), implying that the spermatogonia did not undergo apoptosis.

Antibodies to the MVH protein label all stages of spermatogonia and spermatocytes[36], and PLZF is specifically expressed in the undifferentiated spermatogonia[37]. In the literature, spermatogonia expressing only MVH but not PLZF (MVH^+ PLZF^−) around the basement membrane of seminiferous tubules in the testis are regarded as differentiating spermatogonia[38]. Thus, we simultaneously determined the expression of PLZF and MVH in the cells around the basement membrane of seminiferous tubules in P10 testes of control and *Bcas2^{F/−}*;*Vasa-Cre* mice. Compared with the control, the number of all stages of germ cells (MVH^+) around the basement membrane and differentiating spermatogonia (MVH^+ PLZF^−) was not significantly different in *Bcas2^{F/−}*;*Vasa-Cre* testes (Supplementary Fig. 4a,b). Taken together, these data suggest that the spermatogonia are almost unaffected in *Bcas2^{F/−}*;*Vasa-Cre* mice.

**BCAS2 modulates pre-mRNA splicing in spermatogenesis.** To investigate the molecular consequences of BCAS2 depletion in germ cells, we isolated mRNA from control and *Bcas2^{F/−}*; *Vasa-Cre* testes at P9 and performed RNA sequencing. A total of

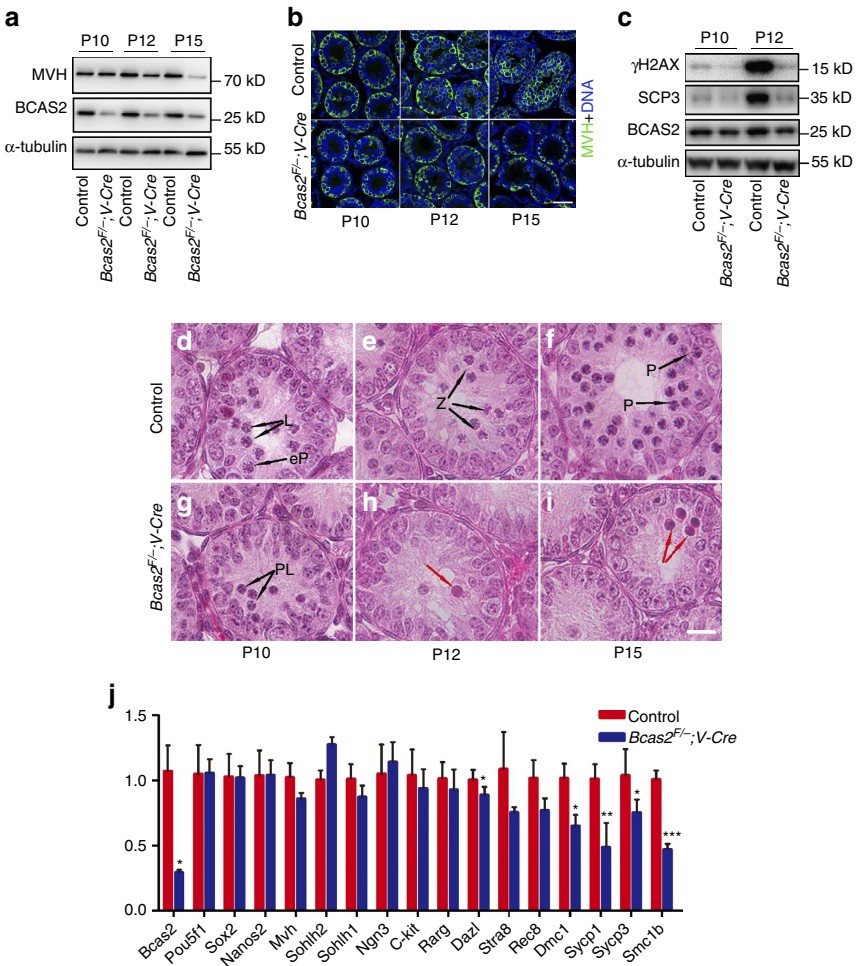

**Figure 3 | Spermatogenesis could not progress into meiosis in Bcas2$^{F/-}$;Vasa-Cre testes.** (**a**) Western blotting analysis of MVH in control and Bcas2$^{F/-}$;Vasa-Cre testes at P10, P12 and P15. α-tubulin was used as the loading control. (**b**) IF staining of MVH in control and Bcas2$^{F/-}$;Vasa-Cre testes at P10, P12 and P15. The DNA was stained with Hoechst 33342. Scale bar, 50 μm. (**c**) Western blotting analyses of SCP3 and γH2AX in control and Bcas2$^{F/-}$;Vasa-Cre testes at P10 and P12. α-tubulin was used as the loading control. (**d–i**) Hematoxylin and eosin (H&E) staining of control (**d–f**) and Bcas2$^{F/-}$;Vasa-Cre testes (**g–i**) at P10, P12 and P15. Spermatogenic cells were shown in cross-sections of seminiferous tubules from control and Bcas2$^{F/-}$;Vasa-Cre testes. Scale bar, 15 μm. Black arrows indicate the representative stages of the spermatocytes. L, leptotene; eP, early pachytene spermatocytes; Z, zygotene spermatocytes; P, pachytene spermatocytes; PL, pre-leptotene spermatocytes; red arrows, apoptotic cells. (**j**) Real-time RT-PCR analysis of marker gene expression in pre-meiotic testes from control and Bcas2$^{F/-}$;Vasa-Cre males at P9 with Gapdh as the internal control. (*$P < 0.05$; **$P < 0.01$; ***$P < 0.001$, $n = 5$). Error bars represent s.e.m.

50619844 and 49955187 clean reads were used for downstream bioinformatics analysis for the control and Bcas2$^{F/-}$;Vasa-Cre testes, respectively (Supplementary Table 1). Paired-end clean reads were aligned to the mouse reference genome using TopHat v2.0.9 software[39]. A total of 46,333,767 (91.55%) and 45,740,076 (91.56%) mapped reads were generated for the control and Bcas2$^{F/-}$;Vasa-Cre samples (Supplementary Table 1). RNA-Seq analyses identified only seven downregulated genes in Bcas2$^{F/-}$;Vasa-Cre testes compared with the control using a common parameter ($P < 0.005$, fold change > 2) (Supplementary Table 2).

To obtain more comprehensive information, we then filtered these data at a lower parameter ($P < 0.05$, fold change > 1.5) to identify more genes potentially regulated by BCAS2 (ref. 40). With this criterion, we found that 59 genes were differentially expressed, among which 20 genes were up-regulated and 39 were down-regulated in Bcas2$^{F/-}$;Vasa-Cre testes (Fig. 5a; Supplementary Table 3). Interestingly, we found that three tubulin genes (Tuba3a, Tuba3b and Tubb4b) were down-regulated. Because BCAS2 has been shown to regulate pre-

mRNA splicing of α-tubulin and γ-tubulin[26,41], we then examined the splicing of these three tubulins by measuring their levels of mature mRNA and pre-mRNA with real-time RT-PCR. Compared with controls, their pre-mRNAs levels were similar, but their mature mRNA levels were significantly down-regulated in Bcas2$^{F/-}$;Vasa-Cre testes (Fig. 5b). Thus, the deletion of BCAS2 perturbs the pre-mRNA splicing of three tubulin genes in mouse testes, suggesting that BCAS2 may be involved in pre-mRNA splicing in mouse spermatogenesis.

Seven major modes of alternative splicing (AS) are described in metazoan organisms, including skipped exons, alternative 5′ splice sites, alternative 3′ splice sites, mutually exclusive exons, alternative first exon, alternative last exon and retained intron[42]. To dissect the impacts of Bcas2 deletion on AS, we compared the AS types between control and Bcas2$^{F/-}$;Vasa-Cre sample based on junction reads and exon expression using Alternative Splicing Detector (ASD) software, a Java program that can assess significant differences in AS events between samples[43]. Compared with the control, 279 AS events were identified as significantly affected (adjusted $P$ value < 0.05) in Bcas2$^{F/-}$;Vasa-Cre sample

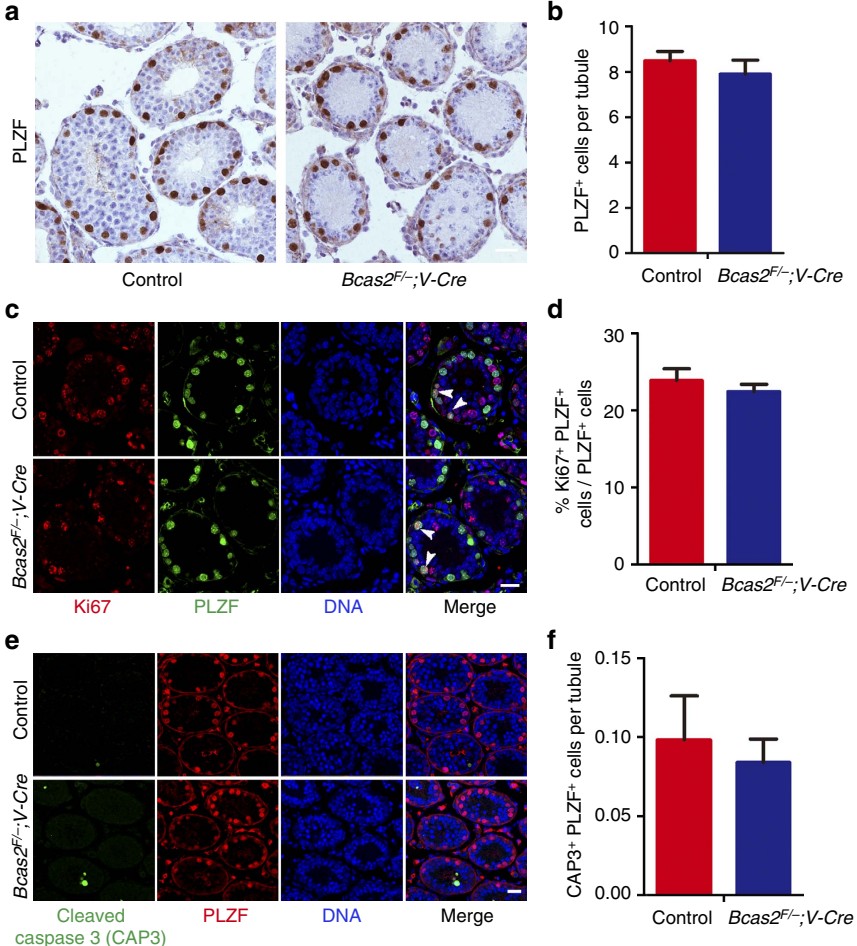

**Figure 4 | Proliferation and apoptosis assay of spermatogonia.** (**a**) Immunohistochemical assay of PLZF in control and *Bcas2^F/−^;Vasa-Cre* testes at P12. The DNA was stained with hematoxylin. Scale bar, 25 μm. (**b**) The number of PLZF$^+$ cells per seminiferous tubules in control and *Bcas2^F/−^;Vasa-Cre* testes at P12. At least 500 tubules were counted from at least 5 different mice. Error bars represent s.e.m. (**c**) IF staining of the mitosis marker Ki67 in PLZF positive cells of control and *Bcas2^F/−^;Vasa-Cre* testes at P12. The DNA was stained with Hoechst 33342. Scale bar, 20 μm. Arrowheads indicate the representative Ki67$^+$ PLZF$^+$ cells. (**d**) The ratio of Ki67$^+$ PLZF$^+$ positive cells in PLZF$^+$ cells of control and *Bcas2^F/−^;Vasa-Cre* testes at P12 (in %). At least 300 tubules were counted from at least 5 different mice. Error bars represent s.e.m. (**e**) IF staining of the apoptosis marker cleaved caspase 3 (CAP3) in PLZF$^+$ cells of control and *Bcas2^F/−^;Vasa-Cre* testes at P12. The DNA was stained with Hoechst 33342. Scale bar, 50 μm. (**f**) The number of CAP3$^+$ PLZF$^+$ cells per seminiferous tubules of control and *Bcas2^F/−^;Vasa-Cre* testes at P12 (in %). At least 150 tubules were counted from 3 different mice. Error bars represent s.e.m.

from 70,786 splicing events (Fig. 5c; Supplementary Table 4). Among the 279 affected AS events, the majority of the splicing events (151) were exon-skipping. In addition, 19 splicing events categorized to alternative 3′ splice sites; 26 to alternative 5′ splice sites; 19 to intron retention; 24 to alternative last exon; 23 to alternative first exon; 12 to mutually exclusive exon; and 5 complicated splicing events to an 'undefined' category (Fig. 5c). These data support the conclusion that BCAS2 is involved in mRNA splicing during mouse spermatogenesis.

**BCAS2 regulates mRNA splicing of functional genes.** From the 279 significantly affected AS events, 245 genes were identified and further analysed for Gene ontology (GO) term enrichment using the DAVID (Database for Annotation, Visualization and Integrated Discovery) software program (Supplementary Table 4). This analysis revealed that 11 genes were involved in sexual reproduction, including 6 genes (*Hsf1*, *Dazl*, *Cit*, *Ehmt2*, *Hmga1* and *Bcl2l11*) related to spermatogenesis (Fig. 5d). We analysed AS changes in these six genes according to the splicing site predicted by ASD software using Integrative Genomics Viewer (IGV), a visualization tool for efficient and

flexible exploration of the large and complicated data sets obtained from sequencing[44]. *Dazl*, *Hsf1* and *Ehmt2* were within the top statistically significant events for the skipped exon category. *Hmga1* and *Bcl2l11* belonged to the alternative first exon category, and *Cit* was categorized to alternative 5′ splice sites. In addition, intron 7 of *Dazl* was significantly retained in *Bcas2^F/−^;Vasa-Cre* testes (Fig. 6a).

We next verified the aberrant splicing of these six genes by semi-quantitative and real-time RT-PCR with specific primers of target genes. We successfully verified the aberrant splicing patterns in *Dazl*, *Ehmt2* and *Hmga1* in *Bcas2^F/−^;Vasa-Cre* testes (Fig. 6b,c). These data suggest that BCAS2 is involved in pre-mRNA splicing of functional genes during mouse spermatogenesis.

**Depletion of BCAS2 results in decreases in DAZL protein.** *Dazl*, encoding a RNA binding protein, is an essential gene for germ cell survival and serves as the intrinsic meiosis-promoting factor during meiosis initiation[45,46]. Strikingly, *Bcas2* depletion led to an obvious switch from the *Dazl*-FL to the *Dazl*-Δ8 isoform and retained intron 7 in *Bcas2^F/−^;Vasa-Cre* testes (Fig. 6b,c). The dramatically altered splicing of *Dazl* and the similar phenotypes

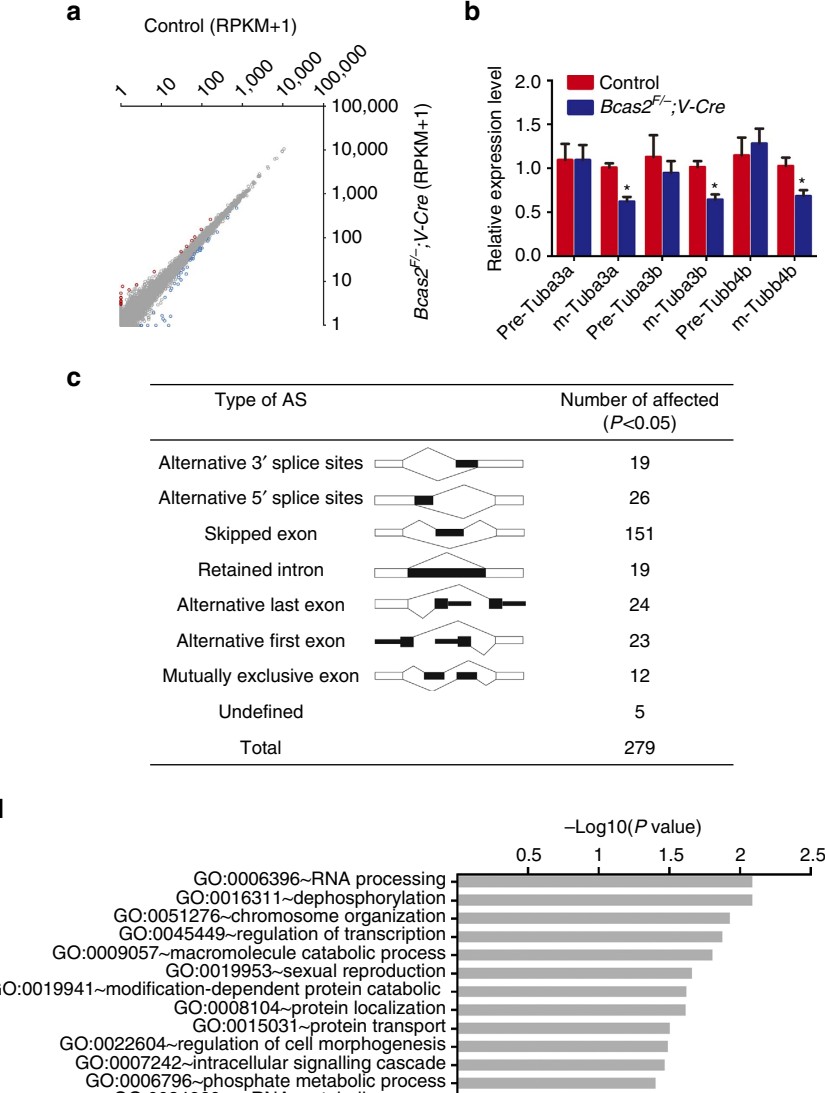

**Figure 5 | Global analysis of *Bcas2*-mediated alternative splicing (AS).** (**a**) Scatter plot of significantly differentially expressed transcripts in *Bcas2^{F/−}*;*Vasa-Cre* testes compared with the controls. Blue dots represent significantly down-regulated transcripts, while red dots show significantly up-regulated transcripts ($P < 0.05$, fold-change of RPKM $> 1.5$). Grey dots illustrated unchanged transcripts. (**b**) Real-time RT-PCR analysis of Pre-mRNA and mature (m) mRNA expression level of *Tuba3a*, *Tuba3b* and *Tubb4b* in control and *Bcas2^{F/−}*;*Vasa-Cre* testes of P9 mice with *Gapdh* as the internal control (*$P < 0.05$, $n = 5$). Error bars represent s.e.m. The pre-mRNA primers were designed in one of the exons and the adjacent intron and the mature mRNA primers were designed to span an exon-exon junction. (**c**) Seven AS events significantly affected by depletion of BCAS2 in the testes at P9. The simple diagrams of seven AS events recognized by ASD software and splicing events affected by depletion of BCAS2 analysing the RNA-seq data using ASD software ($P < 0.05$). (**d**) GO term enrichment analysis of genes with significantly affected AS events.

between *Bcas2^{F/−}*;*Vasa-Cre* and *Dazl* null mice prompted us to investigate the expression of DAZL protein in *Bcas2^{F/−}*;*Vasa-Cre* testes.

Consistent with their mRNA, the proteins of the two isoforms were present in normal testes. The expression of DAZL-FL was much stronger than the expression of the shorter *Dazl*-Δ8. Compared with the controls, the expression of DAZL-FL was dramatically reduced in *Bcas2^{F/−}*;*Vasa-Cre* testes, while DAZL-Δ8 expression was upregulated in *Bcas2^{F/−}*;*Vasa-Cre* testes (Fig. 6d). The DAZL-FL protein was reduced to ~90% in *Bcas2^{F/−}*;*Vasa-Cre* testes at P9 and P12 (Fig. 6e). Meanwhile, DAZL total protein (including DAZL-FL and DAZL-Δ8) was much lower in the *Bcas2^{F/−}*;*Vasa-Cre* testes at P9 and P12 mice compared with the control (Fig. 6f). Thus, these results suggest that BCAS2 may be involved in the splicing of *Dazl*, a germ cell intrinsic factor promoting meiosis initiation.

## Discussion

The alternative pre-mRNA splicing process is implemented by a synergism of the spliceosome, cis-acting RNA elements and specific trans-acting factors[47,48]. The spliceosome is a large and dynamic RNA–protein complex that consists of five small nuclear ribonucleoprotein particles (snRNPs) (U1, U2, U4, U5 and U6) and several hundred proteins that are critical for the recognition of splice sites, assembly of specific stages of the spliceosome and catalytic activity of the splicing reaction[49]. During the mitotic-to-meiotic transition in mouse spermatogenesis, the expression of many key alternative splicing regulators is stage-specific[9]. In addition, proteomic analyses have shown that 58 RNA splicing proteins, including spliceosome components U1–U6 snRNPs-related proteins and many splicing factors, were more highly expressed in type A spermatogonia and pachytene spermatocyte cell clusters[50]. Consistently, the patterns of alternative splicing are

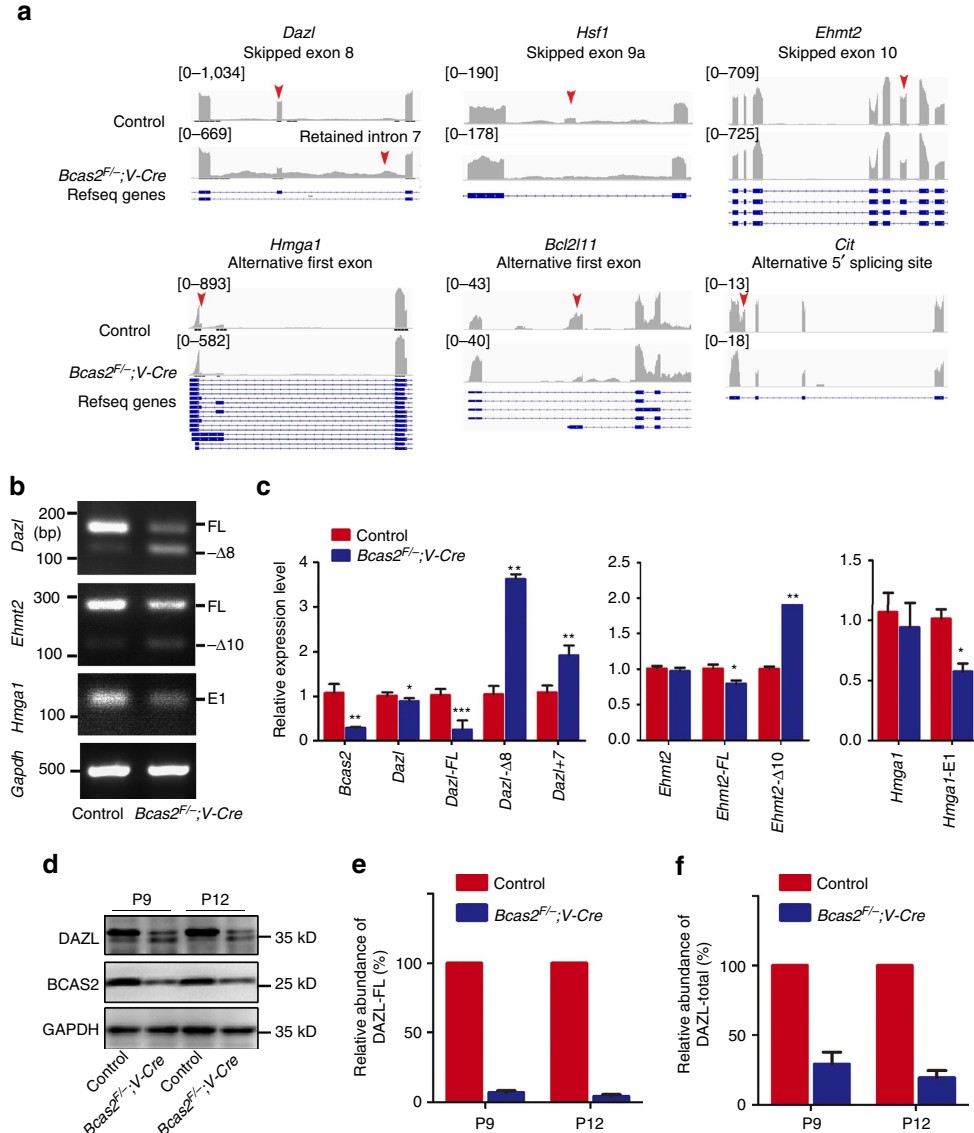

**Figure 6 | *Bcas2* is involved in pre-mRNA splicing of functional genes during spermatogenesis.** (**a**) RNA-seq results of alternative sites in genes related to spermatogenesis using IGV software. Red arrowheads indicate splicing sites. (**b**) RT-PCR analysis of alterative splicing patterns of the changed splicing genes in control and *Bcas2^F/−^;Vasa-Cre* testes of P9 mice with *Gapdh* as the internal control. RT-PCR was performed with specific primers (Supplementary Table 6) in three independent experiments. (**c**) Real-time RT-RCR verified the changed splicing genes with *Hprt* as the internal control. *Dazl*-FL and *Ehmt2*-FL represent the full-length isoform and *Dazl*-Δ8 and *Ehmt2*-Δ10 denote the short form that lacked exon 8 and exon 10, respectively. An alternative first exon of *Hmga1* was shown by *Hmga1*-E1. *Dazl* + 7 primers specifically recognized intron 7 of *Dazl*. *Dazl*, *Ehmt2* and *Hmga1* were detected using primers at the common region (except for the splicing site) of both isoforms (*$P < 0.05$; **$P < 0.01$; ***$P < 0.001$, $n > 5$). Error bars represent s.e.m. (**d**) Western blotting analysis of the expression of two isoforms of DAZL in control and *Bcas2^F/−^;Vasa-Cre* testes of P9 and P12 mice with GAPDH as the loading control. (**e** and **f**) Relative abundances of DAZL-FL (**e**) and DAZL-total (**f**) in control and *Bcas2^F/−^;Vasa-Cre* testes from P9 and P12 mice were determined by Western blotting analyses of four independent experiments. Error bars represent s.e.m.

substantially changed in a stage-specific manner during this process[9]. The stage-enriched expression of splicing proteins and the substantial changes in alternative splicing patterns around meiosis suggest that alternative splicing might be critical for the mitotic-to-meiotic transition during mouse spermatogenesis.

As a core component of the splicing-related Prp19 complex, direct interaction of BCAS2 with PRP19 and the spliceosome is important for spliceosome assembly in yeast[22,51,52]. BCAS2 is universally expressed in organisms, and conventional disruption of BCAS2 results in early embryonic lethality[26,30]. The early embryonic lethality and the complexity of cell types at this stage impede the investigation of BCAS2 in detail. In this study, BCAS2

was comparatively enriched in mouse spermatogonia, prompting us to investigate the role of BCAS2 in mouse spermatogenesis. Using a conditional knockout mouse model, we demonstrated that specific deletion of *Bcas2* in male germ cells leads to a failure of spermatogenesis and male infertility. In *Bcas2*-depleted testes, meiosis failed to initiate and very few meiotic prophase spermatocytes were observed. However, spermatogonia appeared to be grossly normal in *Bcas2^F/−^;Vasa-Cre* male mice. These results suggest that the major defect is most likely in the mitosis-to-meiosis transition during spermatogenesis in *Bcas2^F/−^;Vasa-Cre* male testes. Thus, *Bcas2^F/−^;Vasa-Cre* male mice are a good model for investigating the role of BCAS2 involved in splicing in the mitosis-to-meiosis transition of mouse spermatogonia.

The switch from mitosis to meiosis is a critical step of germ cell development that requires *Dazl*, the 'meiosis promoting factor'[46,53,54]. RNA-binding protein DAZL can target many crucial mRNAs (such as *Mvh*, *Sycp3*, *Tpx-1* and *Tex19.1*) and regulate the translation of these target genes during germ cell development[55–57]. Although the role of its variants is still unclear, *Dazl-FL* and *Dazl-Δ8* variants have been identified in ES cells[58]. Here, we found that the level of DAZL-FL protein is much higher than DAZL-Δ8 in normal mouse testes, suggesting that DAZL-FL might be the main functional form during spermatogenesis. Deletion of *Bcas2* leads to dramatic skipping of exon 8 and significant retention of intron 7 in *Dazl*, suggesting a critical role for the precise control of *Dazl* splicing during normal spermatogenesis. Although the roles of *Dazl-Δ8* with skipped exon 8 and retention of intron 7 are unknown, in fact, we observed a dramatic decrease in DAZL-FL protein and a significant reduction in total DAZL protein, suggesting that deletion of *Bcas2* leads to a significant degree of loss-of-function of DAZL. Although spermatogonia remain in the seminiferous tubules of $Bcas2^{F/-}$;*Vasa-Cre* testes, only a few spermatocytes are observed that do not move into pachytene, which is consistent with observations in *Dazl* null testes[59]. In addition, the protein levels of SCP3, MVH and STRA8, which are targets of DAZL, are dramatically lower in the testes of $Bcas2^{F/-}$;*Vasa-Cre* males. Consistently, the meiotic events (recombination and synapsis) do not occur in $Bcas2^{F/-}$;*Vasa-Cre* testes and the expression of marker genes of meiotic prophase is significantly reduced as early as P9. These observations suggest that *Bcas2* might be is critical for meiotic initiation via regulating *Dazl* splicing.

We also find that *Bcas2* depletion affects splicing of several hundred genes. The affected genes include genes of the functional class 'RNA processing', 'chromosome organization' and 'regulation of transcription'. Importantly, several verified genes are grouped in the functional class 'sexual reproduction'. Many of these genes play important roles in mouse spermatogenesis. *Ehmt2* (also known as *G9a*) is an important mammalian H3K9 methyltransferase and is essential for mouse embryogenesis by transcriptional silencing[60]. *G9a* deficient male germ cells are arrested at the early pachytene stage of meiosis prophase with disordered progression of synaptonemal complex formation[61]. *Hmga1* (the high mobility group A1) is critical for determining chromatin structure and is involved in transcriptional regulation. The tubules of $Hmga1^{+/-}$ male mice are already apparently devoid of spermatocytes and no other stages of spermatozoa and spermatids[62]. Thus, abnormalities in *Ehmt2* and *Hmga1* splicing regulated by *Bcas2* may play a role in both male fertility and growth retardation of the germ cells. In this study, we also found that the abundance of mature *tuba3a*, *tuba3b* and *tubb4b* mRNAs was lower in the $Bcas2^{F/-}$;*Vasa-Cre* testes. Previous studies showed that loss-of-function of CDC5 (core component of Prp19 complex) leads to cell cycle arrest at the G2/M phase because of inefficient splicing of the tubulin-encoding *TUB1* mRNA in *Saccharomyces cerevisiae*[41]. Moreover, microtubule dynamics function in determining Sertoli cell shape, and mitotic and meiotic spindle and sperm flagella development and are thus essential for male fertility[63]. Taken together, these abnormal splicing events resulting from *Bcas2* depletion may also account for the loss of spermatocytes and failure of meiosis initiation.

BCAS2 has multiple functions in several pathways, including RNA splicing and DNA damage repair. However, we did not observe an accumulation of γH2AX expression in spermatogonia in $Bcas2^{F/-}$;*Vasa-Cre* testes during the meiosis initiation phase (Supplementary Fig. 5). Furthermore, the expression of most genes is not affected in $Bcas2^{F/-}$;*Vasa-Cre* testes at P9. These data suggest that BCAS2 mainly functions through RNA splicing but not DNA damage in mouse spermatogenesis.

In conclusion, our results suggest that BCAS2 is involved in pre-mRNA splicing in spermatogonia and is essential for mouse spermatogenesis and male fertility. This research provides the first evidence for alternative splicing machinery regulating meiosis initiation in germ cells during mouse spermatogenesis.

## Methods

**Mice maintenance and generation of gene-targeted mice.** All mice were maintained under specific-pathogen-free (SPF) conditions in compliance with the guidelines of the Animal Care and Use Committee of the Institute of Zoology at the Chinese Academy of Sciences (CAS). The generation of the $Bcas2^{Floxed/Floxed}$ ($Bcas2^{F/F}$) mouse line was described previously[30]. Male $Bcas2^{F/F}$ mice were crossed with *Vasa-Cre* transgenic mice to obtain mice with *Bcas2* specifically ablated in the male germ line. The Bcas2$^{F/F}$;*Vasa-Cre* mouse line was maintained on a mixed background (129/C57BL/6). Genotyping of *Bcas2* was performed by PCR of mice tail genomic DNA. Forward primer 1 (F1, 5-ATTCCAGCAGTTGGTGTGGG-3) and a reverse primer (R, 5-CATTGCTGGACAGAAGGTGAG-3) were used to detect the wild-type allele (402 bp) and the floxed allele (522 bp), and forward primer 2 (F2, 5-AGGTGTATGAATGCCTGAACAAG-3) and reverse primer R were used to detect the deleted allele (394 bp) in PCR genotyping. *Vasa-Cre* was genotyped with the forward primer (F-V, 5-CACGTGCAGCCGTTTAAGCCGC GT-3) and a reverse primer (R-V, 5-TTCCCATTCTAAACAACACCCTGAA-3) to produce a 240 bp product. $Bcas2^{F/-}$ male mice were used as controls and $Bcas2^{F/-}$;*Vasa-Cre* male mice were used as mutants.

**Histological analysis, immunostaining and imaging.** Testes from control and $Bcas2^{F/-}$;*Vasa-Cre* male mice were isolated and fixed in Bouin's solution (Saturated picric acid: 37% Formaldehyde: Glacial acetic acid = 15: 5: 1) overnight at room temperature for histological analysis and in 4% paraformaldehyde (PFA) overnight at 4 °C for immunostaining. The samples were dehydrated stepwise through an ethanol series (30, 50, 70, 80, 90 and 100% ethanol), embedded in paraffin, and sectioned (5 μm). After dewaxing and hydration, the sections were stained with hematoxylin and 1% eosin and imaged with a Nikon ECLIPSE Ti microscope.

For immunostaining, following dewaxing and hydration, the sections were boiled in Buffer TE (10 mM Tris, 1 mM EDTA, pH 9.0) for 20 min using a microwave oven for antigen retrieval. After washing with PBS 3 times, the sections were permeated in 0.2% Triton X-100 for 15 min, blocked with 5% BSA in PBS for 1 h at room temperature and incubated overnight at 4 °C with primary antibodies diluted in 1% BSA. Following three washes with PBS, secondary antibodies were added to the sections and incubated for 1 h at room temperature. The sections were then washed in PBS three times, incubated in 2 μg ml$^{-1}$ of Hoechst 33342 (Invitrogen, H21492) for 10 min at room temperature, and mounted with Fluoromount-G medium (Southern Biotech, 0100-01).

The primary antibodies were as follows: rabbit anti-BCAS2 polyclonal antibody (10414-1-AP, Proteintech, 1:200); mouse anti-PLZF monoclonal antibody (sc28319, Santa Cruz, 1:200); rabbit anti-DDX4/MVH polyclonal antibody (ab13840, Abcam, 1:250); mouse phosphor-Histone H2A.X (Ser139/Tyr142) antibody (#5438, CST, 1: 500); mouse anti-SCP3 monoclonal antibody (sc74569, Santa Cruz, 1:200); rabbit anti-Stra8 polyclonal antibody (ab49405, Abcam, 1:250); rabbit anti-Cleaved Caspase-3 (Asp175) antibody ( #9661, Cell Signaling, 1:200); and rabbit anti-Ki67 antibody (ab92742, Abcam, 1:400).

The secondary antibodies were as follows: For immunohistochemistry of PLZF, horseradish peroxidase-goat anti-mouse IgG (1:500, Jackson ImmunoResearch) was incubated for 1 h at room temperature. For immunofluorescence staining, Alexa Fluor 488 donkey anti-rabbit (Jackson, 1:1,000); Alexa Fluor 549 donkey anti-rabbit (Jackson, 1:1,000); Alexa Fluor 488 Donkey Anti-Mouse (Jackson, 1:200); and Alexa Fluor 549 donkey anti-Mouse (Jackson, 1:200) were used.

The immunohistochemistry images of PLZF were obtained under the bright-field of a Nikon ECLIPSE Ti microscope. The immunofluorescence staining was imaged with a laser scanning confocal microscope LSM780 (Carl Zeiss).

**RNA extraction and real-time RT-PCR.** Total RNA was extracted from whole testes using Trizol reagent (Invitrogen, 15596-026) following the manufacturer's instructions. After removing the residual genomic DNA with the DNase I Kit (Promega, M6101), 500 ng of total RNA was reverse-transcribed into cDNAs using the PrimeScript RT Reagent Kit (TaKaRa, RR037A) according to the manufacturer's protocol. Real-time RT-PCR was performed using a SYBR Premix Ex Taq kit (TaKaRa, DRR420A) on a LightCycler 480 instrument (Roche). Relative gene expression was analysed based on the $2^{-\Delta\Delta Ct}$ method with *Hprt* and *Gapdh* as internal controls.

Primers used to determine the amount of *Tuba3a*, *Tuba3b* and *Tubb4b* pre-mRNA were targeted to the 3′ end of exon3 and the neighbouring 5′ end of intron 3. Primers used to determine the abundance of mature mRNAs of *Tuba3a*, *Tuba3b* and *Tubb4b* were designed to span an exon-exon junction, with *Tuba3a*-mRNA primers annealed to the 3′ end of exon4 and 5′end of exon5, *Tuba3b*-mRNA primers annealed to the 3′ end of exon1 and 5′end of exon2 and *Tubb4b*-mRNA primers annealed to the 3′ end of exon2 and 5′end of exon3.

The information of all primers was listed in Supplementary Table 5.

**Western blotting.** Protein samples were prepared using RIPA lysis buffer (50 mM Tris–HCl (pH 7.5), 150 mM NaCl, 1% Sodium deoxycholate, 1% Triton X-100, 0.1% SDS, 5 mM EDTA, 1 mM $Na_3VO_4$, 5–10 mM NaF) containing a protease inhibitor cocktail (Roche, 04693132001) and quantified using a BCA reagent kit (Beyotime, P0012-1). Equal amounts of total protein were separated in a 10% SDS–PAGE gel and transferred onto PVDF membranes. After blocking with 5% non-fat milk for 1 h at room temperature, the membranes were incubated with diluted primary antibodies at 4 °C overnight. After three washes with TBST, the membranes were incubated with secondary antibodies conjugated with horseradish peroxidase (1:3,000, Jackson ImmunoResearch) at room temperature for 1 h. The signals were developed with Pierce ECL Substrate (Thermo Fisher Scientific, #34080), detected with Bio-RAD ChemiDocTMXRs $^+$ and analysed with Quantity One software (Bio-Rad Laboratories). All uncropped western blots can be found in Supplementary Fig. 6.

The primary antibodies used were as follows: rabbit anti-BCAS2 polyclonal antibody (10414-1-AP, Proteintech, 1:1000); rabbit anti-PLZF antibody (sc22839, Santa Cruz, 1:1,000); rabbit anti-DDX4/MVH polyclonal antibody (ab13840, Abcam, 1:1,000); mouse phosphor-Histone H2A.X (Ser139/Tyr142) antibody (#5438, CST, 1:2,000); rabbit anti-H2AX (phospho S139) antibody (ab22551, Abcam, 1:500); rabbit anti-SCP3 antibody (ab15093, Abcam, 1:1,000); rabbit anti-Stra8 polyclonal antibody (ab49405, Abcam, 1:500); rabbit anti-α-Tubulin antibody (#2144, CST, 1:1,000); anti-GAPDH (1C4) mouse mAb (KM9002, Sungene Biotech, 1:3,000); and anti-DAZL mouse antibody (MCA2336, AbD Serotec, 1:1,000).

**Enrichment of spermatogenic cells and flow cytometry analysis.** The spermatogenic cells and somatic cells were enriched using a two-step enzymatic digestion process followed by a differential adhesion method as previously described with some modifications[31,64]. Briefly, after the tunica albuginea was disrupted, testes from P9 wild-type ICR mice were transferred into 2 mg ml $^{-1}$ type IV collagenase (Sigma, C5138) and incubated at 37 °C for 20 mins with gentle shaking every 3–5 min to accelerate the testes dissociation. Then, the cell suspension was centrifuged at 1000 r.p.m. for 3 mins and the pellet was digested with 0.25% Trypsin-EDTA (Gibco, 25200-072) at 37 °C for 10 min to dissociate the seminiferous tubules into single cells. The suspension was neutralized with 5 ml DMEM supplemented with 10% FBS and centrifuged at 1000 r.p.m. for 3 min. The cell pellet was suspended in 8 ml of ES medium and seeded in a 10 cm culture dish. After 2–3 h of incubation at 37 °C, the floating and weakly adhering cells were transferred to a new 10 cm dish. The fraction of spermatogenic cells (FSPCs) was the floating and weakly adhering cells. The attached cells on the bottom of the dish were collected as the fraction of somatic cells (FSCs). The efficiency of separation was examined using RT-PCR and flow cytometry analysis.

For flow cytometry analysis, the separated cells were fixed in 4% PFA for 5 min. After washing with PBS, the cells were permeated in 0.5% Triton X-100 for 5 min, blocked with 5% BSA in PBS for 1 h at room temperature. The experimental groups were incubated overnight at 4 °C with MVH antibody (ab13840, Abcam, 1:200) diluted in 1% BSA and the negative control group was incubated overnight at 4 °C with 1% BSA. Following three washes with PBS, Alexa Fluor 488 donkey anti-rabbit secondary antibodies (Jackson, 1:1,000) were incubated for 1 h at room temperature. FITC signals of MVH positive cells were detected by flow cytometry and the results were analysed by Flowjo software (Tree star).

**RNA sequencing.** Testes samples were collected from 9-day-old control and $Bcas2^{F/-}$;Vasa-Cre mice. Total RNA was isolated using the Trizol reagent according to the manufacturer's protocol and treated with DNase I to remove residual genomic DNA. The purity, concentration and integrity were assessed using a NanoPhotometer spectrophotometer (IMPLEN), Qubit RNA Assay Kit in Qubit 2.0 Fluorometer (Life Technologies) and Nano 6,000 Assay Kit of the Bioanalyzer 2,100 system (Agilent Technologies), respectively. A total amount of 2 μg of RNA per sample was used to prepare cDNA libraries, which were generated using the NEBNext Ultra RNA Library Prep Kit for Illumina (NEB) following the manufacturer's recommendations. For each cDNA library, 5G base pairs (raw data) were generated by Illumina Hi-Seq 2500. After base composition and quality tests were passed, we removed the adaptor sequence, and sequences with a high content of unknown bases (unknown bases more than 10%) or low quality reads. The clean reads were used for bioinformatic analysis.

**Bioinformatic analyses.** Paired-end clean reads were mapped to the mouse reference genome using Tophat2 v2.0.9 software according to the standard protocol. The number of mapped reads were counted using HTSeq v[0.6.1]. Differential expression analysis of control and $Bcas2^{F/-}$;Vasa-Cre samples was performed using the DEG Seq R package (1.12.0). The resulting P-values were adjusted using Benjamini and Hochberg's approach for controlling the false discovery rate. First, we used fold change > 2 and P value < 0.005 as the threshold to filter for significantly different expression. Then, we used the lower threshold (fold change > 1.5 and P value < 0.05) to filter for more significantly different expression.

AS patterns were analysed using ASD (AS detector) software with the standard protocol[43]. Briefly, the '.bam' files generated from the RNA-seq data after mapping using Tophat2 software were analysed by the ASD software for AS analysis. The ASD software identified the seven common modes of AS events for control and $Bcas2^{F/-}$;Vasa-Cre samples and obtained an adjusted P value difference between control and $Bcas2^{F/-}$;Vasa-Cre samples by combining two P values calculated using the Fisher exact test (P value of junction read-counts and P value based on the alternative exon read coverage relative to its gene read coverage between control and $Bcas2^{F/-}$;Vasa-Cre samples) using a weighted arithmetic equation. The GO term enrichment and functional annotation analyses of genes identified by ASD software were conducted using database for annotation, visualization and integrated discovery (DAVID). The integrative genomics viewer tool (IGV) was used for efficient and flexible visualization and exploration of spliced sites between control and $Bcas2^{F/-}$;Vasa-Cre samples on standard desktop computers.

**Statistical analysis.** The results of all quantitative experiments were based on at least three independent biological samples. Statistical analysis was performed by using at least five samples and expressed as the mean ± s.e.m. The results of statistical analyses were subjected to a Student's two-tailed t-test with a significance level of P < 0.05. Equal variances were not formally tested. No statistical method was used to predetermine sample sizes; however, sample sizes used in current study were similar as previously reported[17,18,30]. Experiments were not randomized.

**Data availability.** Data for RNA sequencing of P9 testes from control and $Bcas2^{F/-}$;V-Cre males have been deposited in the Gene Expression Omnibus database under accession code GSE89801. The authors declare that all data supporting the findings of this study are available within the article and its supplementary information files or from the corresponding author on reasonable request.

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

# ARTICLE

21. Ajuh, P. *et al.* Functional analysis of the human CDC5L complex and identification of its components by mass spectrometry. *EMBO J.* **19,** 6569–6581 (2000).

22. Chan, S. P., Kao, D. I., Tsai, W. Y. & Cheng, S. C. The Prp19p-associated complex in spliceosome activation. *Science* **302,** 279–282 (2003).

23. Bessonov, S., Anokhina, M., Will, C. L., Urlaub, H. & Luhrmann, R. Isolation of an active step I spliceosome and composition of its RNP core. *Nature* **452,** 846–U843 (2008).

24. Chen, H. R. *et al.* Snt309p, a component of the Prp19p-associated complex that interacts with Prp19p and associates with the spliceosome simultaneously with or immediately after dissociation of U4 in the same manner as Prp19p. *Mol. Cell Biol.* **18,** 2196–2204 (1998).

25. Chen, H. R. *et al.* Snt309p modulates interactions of Prp19p with its associated components to stabilize the Prp19p-associated complex essential for pre-mRNA splicing. *Proc. Natl Acad. Sci. USA* **96,** 5406–5411 (1999).

26. Chen, P. H. *et al.* BCAS2 is essential for *Drosophila* viability and functions in pre-mRNA splicing. *RNA–Publ. RNA Soc.* **19,** 208–218 (2013).

27. Chou, M. H. *et al.* BCAS2 regulates delta-notch signaling activity through delta pre-mRNA splicing in *Drosophila* wing development. *PLOS ONE* **10,** e0130706 (2015).

28. Wan, L. & Huang, J. The PSO4 protein complex associates with Replication Protein A (RPA) and modulates the function of ataxia telangiectasia-mutated and Rad3-related (ATR). *J. Biol. Chem.* **289,** 6619–6626 (2014).

29. Marechal, A. *et al.* PRP19 transforms into a sensor of RPA-ssDNA after DNA damage and drives ATR activation via a ubiquitin-mediated circuitry. *Mol. Cell* **53,** 235–246 (2014).

30. Xu, Q. H. *et al.* Maternal BCAS2 protects genomic integrity in mouse early embryonic development. *Development* **142,** 3943–3953 (2015).

31. Sato, T., Katagiri, K., Kubota, Y. & Ogawa, T. *In vitro* sperm production from mouse spermatogonial stem cell lines using an organ culture method. *Nat. Protoc.* **8,** 2098–2104 (2013).

32. Gallardo, T., Shirley, L., John, G. B. & Castrillon, D. H. Generation of a germ cell-specific mouse transgenic Cre line, Vasa-Cre. *Genesis* **45,** 413–417 (2007).

33. Handel, M. A. & Schimenti, J. C. Genetics of mammalian meiosis: regulation, dynamics and impact on fertility. *Nat. Rev. Genet.* **11,** 124–136 (2010).

34. Anderson, E. L. *et al.* Stra8 and its inducer, retinoic acid, regulate meiotic initiation in both spermatogenesis and oogenesis in mice. *Proc. Natl Acad. Sci. USA* **105,** 14976–14980 (2008).

35. Ahmed, E. A. & de Rooij, D. G. Staging of mouse seminiferous tubule cross-sections. *Methods Mol. Biol.* **558,** 263–277 (2009).

36. Toyooka, Y. *et al.* Expression and intracellular localization of mouse Vasa-homologue protein during germ cell development. *Mech. Dev.* **93,** 139–149 (2000).

37. Costoya, J. A. *et al.* Essential role of Plzf in maintenance of spermatogonial stem cells. *Nat. Genet.* **36,** 653–659 (2004).

38. Hao, J. *et al.* Sohlh2 knockout mice are male-sterile because of degeneration of differentiating type A spermatogonia. *Stem Cells* **26,** 1587–1597 (2008).

39. Kim, D. *et al.* TopHat2: accurate alignment of transcriptomes in the presence of insertions, deletions and gene fusions. *Genome Biol.* **14,** R36 (2013).

40. Baumer, D. *et al.* Alternative splicing events are a late feature of pathology in a mouse model of spinal muscular atrophy. *PLOS Genet.* **5,** e1000773 (2009).

41. Burns, C. G. *et al.* Removal of a single alpha-tubulin gene intron suppresses cell cycle arrest phenotypes of splicing factor mutations in *Saccharomyces cerevisiae*. *Mol. Cell Biol.* **22,** 801–815 (2002).

42. Shapiro, I. M. *et al.* An EMT-driven alternative splicing program occurs in human breast cancer and modulates cellular phenotype. *PLOS Genet.* **7,** e1002218 (2011).

43. Zhou, X. X. *et al.* Transcriptome analysis of alternative splicing events regulated by SRSF10 reveals position-dependent splicing modulation. *Nucleic Acids Res.* **42,** 4019–4030 (2014).

44. Robinson, J. T. *et al.* Integrative genomics viewer. *Nat. Biotechnol.* **29,** 24–26 (2011).

45. Seligman, J. & Page, D. C. The *Dazh* gene is expressed in male and female embryonic gonads before germ cell sex differentiation. *Biochem. Biophys. Res. Co.* **245,** 878–882 (1998).

46. Lin, Y. F., Gill, M. E., Koubova, J. & Page, D. C. Germ cell-intrinsic and -extrinsic factors govern meiotic initiation in mouse embryos. *Science* **322,** 1685–1687 (2008).

47. Chen, M. & Manley, J. L. Mechanisms of alternative splicing regulation: insights from molecular and genomics approaches. *Nat. Rev. Mol. Cell Biol.* **10,** 741–754 (2009).

48. Fu, X. D. & Ares, M. Context-dependent control of alternative splicing by RNA-binding proteins. *Nat. Rev. Genet.* **15,** 689–701 (2014).

49. Wahl, M. C., Will, C. L. & Luhrmann, R. The spliceosome: design principles of a dynamic RNP machine. *Cell* **136,** 701–718 (2009).

50. Gan, H. *et al.* Integrative proteomic and transcriptomic analyses reveal multiple post-transcriptional regulatory mechanisms of mouse spermatogenesis. *Mol. Cell Proteomics* **12,** 1144–1157 (2013).

51. Grote, M. *et al.* Molecular architecture of the human Prp19/CDC5L complex. *Mol. Cell Biol.* **30,** 2105–2119 (2010).

52. Hogg, R., McGrail, J. C. & O'Keefe, R. T. The function of the NineTeen Complex (NTC) in regulating spliceosome conformations and fidelity during pre-mRNA splicing. *Biochem. Soc. Trans.* **38,** 1110–1115 (2010).

53. Koubova, J. *et al.* Retinoic acid activates two pathways required for meiosis in mice. *PLOS Genet.* **10,** e1004541 (2014).

54. Soh, Y. Q. S. *et al.* A gene regulatory program for meiotic prophase in the fetal ovary. *PLOS Genet.* **11,** e100553 (2015).

55. Reynolds, N., Collier, B., Bingham, V., Gray, N. K. & Cooke, H. J. Translation of the synaptonemal complex component Sycp3 is enhanced *in vivo* by the germ cell specific regulator Dazl. *RNA.* **13,** 974–981 (2007).

56. Reynolds, N. *et al.* Dazl binds *in vivo* to specific transcripts and can regulate the pre-meiotic translation of *Mvh* in germ cells. *Hum. Mol. Genet.* **14,** 3899–3909 (2005).

57. Jenkins, H. T., Malkova, B. & Edwards, T. A. Kinked β-strands mediate high-affinity recognition of mRNA targets by the germ-cell regulator DAZL. *Proc. Natl Acad. Sci. USA* **108,** 18266–18271 (2011).

58. Xu, X. B. *et al.* Mouse Dazl and its novel splice variant functions in translational repression of target mRNAs in embryonic stem cells. *BBA-Gene Regul. Mech.* **1829,** 425–435 (2013).

59. Schrans-Stassen, B. H. G. J., Saunders, P. T. K., Cooke, H. J. & de Rooij, D. G. Nature of the spermatogenic arrest in Dazl$^{-/-}$ mice. *Biol. Reprod.* **65,** 771–776 (2001).

60. Tachibana, M. *et al.* G9a histone methyltransferase plays a dominant role in euchromatic histone H3 lysine 9 methylation and is essential for early embryogenesis. *Gene Dev.* **16,** 1779–1791 (2002).

61. Tachibana, M., Nozaki, M., Takeda, N. & Shinkai, Y. Functional dynamics of H3K9 methylation during meiotic prophase progression. *EMBO J.* **26,** 3346–3359 (2007).

62. Liu, J., Schiltz, J. F., Ashar, H. R. & Chada, K. K. Hmga1 is required for normal sperm development. *Mol. Reprod. Dev.* **66,** 81–89 (2003).

63. O'Donnell, L. & O'Bryan, M. K. Microtubules and spermatogenesis. *Semin. Cell Dev. Biol.* **30,** 45–54 (2014).

64. Kubota, H. & Brinster, R. L. Culture of rodent spermatogonial stem cells, male germline stem cells of the postnatal animal. *Methods Cell Biol.* **86,** 59 (2008).

## Acknowledgements

We thank Shiwen Li for assistance with confocal microscopy, Lianjun Zhang and Min Chen and all members in Li's lab for their helpful advice. The work was funded by the National Natural Science Foundation of China (31590832, 31171382) and the Beijing Municipal Natural Science Foundation (5162005).

## Author contributions

W.L. designed and performed the major experiments and data analysis and wrote the manuscript. F.W. established the *Bcas2$^{Floxed/Floxed}$* mice. Q.X., J.S., X.Z., X.L., Z.Z., Z.G. and H.M. contributed to mouse maintenance and technical assistance. E.D. and F.G. contributed to data analysis. L.L., Z.Y. and S.G. initiated and organized the study, analysed the data and wrote the manuscript. All authors commented on the manuscript.

## Additional information

**Competing financial interests**: The authors declare no competing financial interests.

**How to cite this article**: Liu, W. *et al.* BCAS2 is involved in alternative mRNA splicing in spermatogonia and the transition to meiosis. *Nat. Commun.* **8,** 14182 doi: 10.1038/ncomms14182 (2017).

**Publisher's note**: 

