## [Peer Review File · Nature Communications]

Reviewers' comments:

Reviewer #1 (Remarks to the Author):

What are the major claims of the paper? BCAS2 protein is enriched within spermatogonial germ cells, essential for fertility, and involved in controlling alternative splicing in meiosis. Amongst the target exons identified was a splice variant of DAZL which encodes a critical germ cell translation factor. The authors show convincingly that BCAS2 is not expressed within the vasa-Cre generated KO mice, and that there is a significant germ cell defect. By the use of markers and staining they narrow this defect down to meiotic entry. The authors discover some interesting splice patterns that change, although overall patterns of gene expression were similar in the WT and KO./ Are the claims novel? If not, please identify the major papers that compromise novelty. These claims are novel. There is a lot of interest in the functions of DAZL, do this paper should be of interest widely.

Will the paper be of interest to others in the field? Yes, to scientists interested in both alternative splicing and germ cell development including meiosis.

Will the paper influence thinking in the field? It is very interesting that the BCAS2 protein has such specific target effects on splicing.

Are the claims convincing? If not, what further evidence is needed? The paper provides strong evidence for its conclusions. Yet, I tried some of the primers within in silico PCR on the UCSC genome browser (the DAZL primer pairs) and this did not predict any products, so can the authors check these sequences. In particular, I wanted to know about the DAZL exon was a multiple of 3. Why does the lower band on the western in Figure 7D not change expression to equal the original upper band in the WT? I notice the exon upstream of DAZL exon 8 is an alternative polyA site -is this used at all? The read height of the RNAseq tracks should be shown. How many replicates were used for the RNAseq (Figure 7a) and RT-PCR (Figure 7b). At least the endpoint PCRs should be done in triplicate, and this data should be presented.

Are there other experiments that would strengthen the paper further? How much would they improve it, and how difficult are they likely to be? To prove the role of BCAS2 it would be possible to deplete BCAS2 from ES cells, and then monitor effects on DAZL; or carry out minigene experiments to see if expression of BCAS2 changed DAZL splicing patterns. These would both strengthen the connection between BCAS2 and the identified splicing targets. Having said that, if these experiments did not work they would not disprove the main claims.

Are the claims appropriately discussed in the context of previous literature? Yes

If the manuscript is unacceptable in its present form, does the study seem sufficiently promising that the authors should be encouraged to consider a resubmission in the future? This manuscript seems potentially acceptable.

The paper is well written, the abstract/text appropriate. There are some spelling mistakes, e.g. alternative is spelled wrongly in Figure 7.

Reviewer #2 (Remarks to the Author):

In this study, the authors address gaps in knowledge of the influence that pre-mRNA splicing has in spermatogenesis which is grossly undefined at present. Studying the role of BCAS2 in spermatogenesis is certainly warranted and could yield important new information. Although the phenotype of BCAS2 conditional knockout is interesting, the depth of examination for impaired spermatogenesis is not sufficient to draw definitive conclusions.

Data supporting the claim that BCAS2 is highly expressed in spermatogonia is not convincing and the authors should temper their conclusions. Immunostaining is not a quantitative approach that yields data to support conclusions about the levels of BCAS2 in spermatogonia, other germ cells, or somatic cells. With the level of analysis conducted, statements such as "BCAS2 expression was much stronger in PLZF+ and PLZF- cells" is simply not supported.

Claims of isolating spermatogonia and somatic cells from P9 testes is not validated. At best, the approach used results in enrichment of the cell types. Thus, the more conservative assessments should be made. Also, the label of SSC in the bar graph of Figure 1b is not accurate. The supposed germ cell population that was isolated is a heterogeneous mix of cells.

The RT-PCR methodology for distinguishing between pre-mRNA and mature mRNA for the Tub3 isoforms is not described. Thus, the reader cannot independently assess validity of the approach.

All assessment of disrupted spermatogenesis is made within the context of the first round of spermatogenesis which may be unique and not reflect the role of BCAS2 or alternative splicing in normal spermatogenesis. In mice, the first meicytes that arise in postnatal life around P8 are known to be derived from a subset of prospermatogonial precursors that do not transit through an undifferentiated spermatogonial state. All other spermatocytes produced in steady-state spermatogenesis are produced from spermatogonial stem cells that arose from a different subset of prospermatogonial precursors. Thus, the first round of spermatogenesis and meicytes are unique compared to subsequent populations. In the current study, the authors only examined the first round spermatocytes and the lack of subsequent second round spermatocytes suggests a defect in spermatogonial differentiation. For these reasons, definitive conclusions about the role of BCAS2 or alternative splicing in meiosis cannot be made. Further experimentation into the cause of impaired spermatogenesis is warranted.

Minor Comments

The manuscript contains many typos and grammatical errors. For example, 'alternative' is misspelled multiple time in Figure 7, Western blotting should be capitalized throughout, etc. Incorrect terminology for spermatogenesis is used throughout the manuscript. For example, spermatogonia is often used when the correct term is spermatogonial, and statements indicating that SSCs undergo meiosis and spermiogenesis are not accurate (spermatocytes undergo meiosis and spermatids undergo spermiogenesis).

The authors should use the term prospermatogonia in place of gonocyte.

The title of the manuscript is not accurate. The context is misleading because as written the title indicates the BCAS2 splices spermatogonia, I'm sure the authors mean mRNA splicing here but that is not how the statement is written. In addition, the authors' main conclusion is that BCAS2 and hence alternative splicing influence meiotic progression but this is not reflected in the title.

Reviewer #3 (Remarks to the Author):

Within this manuscript the authors define a role of BCAS2 in mRNA splicing in spermatogonia and ultimately male fertility. This is a novel finding and the paper contains data of high quality. The figures are well presented. I have no major concerns regarding the methodology or the data interpretation. There are a few areas where the text could be clarified and thus help readers interpret the ultimate story however. The first of these is the title - as written the title indicates a fundamental problem in spermatogonial function, while this is technically true, the problems do not manifest until early meiosis. I wonder if 'BCAS2 is (note typo in current draft) involved in alternative mRNA splicing in spermatogonia and the transition to meiosis, and male fertility', would be more informative.

Abstract - that data clearly shows that BCAS2 dysfunction leads to aberrant splicing of many mRNA, several of which could result in sterility. While I agree that mentioning DAZL is informative, the emphasis is distracting. It also begs the question of over expressing DAZL to correct the defect - this would probably not recover fertility.

Minor points - p7 last line. Replace 'stage' with 'type'

The experiments described under the heading 'BCAS2 is critical for germ cell meiosis in mouse testis' needs to be rewritten. At this stage of the manuscript, a second possibility formally exists ie. that spermatogonia are failing to commit to meiosis. Later in the manuscript is possibility is eliminated. Please modify the text at the bottom of p8 to indicate this possibility.

Similarly, the experiments described at the top of p9 are largely an empty analysis. You can see by histology that the cells are missing, so it's no surprise that the markers are all decreased - this could be de-emphasised a little.

p10 - as these cells have an apoptotic morphology (pyknotic nuclei) it is not possible to definitely tell whether they are have just entered the apoptotic pathway (ie. would be pachytene at d15) or they have been arrested for several days prior to becoming apoptotic. The best you can say is an arrest during early prophase I. Please modify the text. 'Period' rather than 'stage'

The point-by-point responses to the reviewers are listed as follows:

Response to Reviewer #1:

We are pleased with the positive comments and appreciate the suggestions to strengthen the manuscript. The point-by-point responses are listed below:

Comments 1: Are the claims convincing? If not, what further evidence is needed? The paper provides strong evidence for its conclusions. Yet, I tried some of the primers within in silico PCR on the UCSC genome browser (the *DAZL* primer pairs) and this did not predict any products, so can the authors check these sequences.

Response to #1: Thanks for your comment. We have carefully checked the sequences of these primers. When *Dazl* primer pairs are input to in silico PCR on the UCSC genome browser, correct products are successfully obtained as follows:

The results obtained using *Dazl* (in the common region) primers for the real-time RT-PCR:

UCSC In-Silico PCR

The sequences and coordinates shown below are from UCSC Genes, not from the genome assembly.

```
>uc008cyv.3\_Dazl:364+457 94bp GGATGAAACCGAAATCAGGA ATAGCCCTTCGACACACCAG  
GGATGAAACCGAAATCAGGAgtttctttgccagatatggctcagtaaaag  
aagtgaagataatcactgatcgaaCTGGTGTGTCGAAGGGCTAT
```

```
>uc033hel.1\_Dazl:364+457 94bp GGATGAAACCGAAATCAGGA ATAGCCCTTCGACACACCAG  
GGATGAAACCGAAATCAGGAgtttctttgccagatatggctcagtaaaag  
aagtgaagataatcactgatcgaaCTGGTGTGTCGAAGGGCTAT
```

```
>uc008cyu.3\_Dazl:222+315 94bp GGATGAAACCGAAATCAGGA ATAGCCCTTCGACACACCAG  
GGATGAAACCGAAATCAGGAgtttctttgccagatatggctcagtaaaag  
aagtgaagataatcactgatcgaaCTGGTGTGTCGAAGGGCTAT
```

```
>uc008cyw.2\_Dazl:364+457 94bp GGATGAAACCGAAATCAGGA ATAGCCCTTCGACACACCAG  
GGATGAAACCGAAATCAGGAgtttctttgccagatatggctcagtaaaag  
aagtgaagataatcactgatcgaaCTGGTGTGTCGAAGGGCTAT
```

The results obtained using *Dazl*-FL primers for the real-time RT-PCR:

UCSC In-Silico PCR

The sequences and coordinates shown below are from UCSC Genes, not from the genome assembly.

```
>uc008cyv.3_Dazl:616+823 208bp TCCACCACAGTTCAGAGTG AACATAACTCCTCTGCTCTCCA  
TCCACCACAGTTCAGAGTGtttgagtagtccaaatgctgagacttaca  
tcagcctccaaccatgatgaatcctatcaactcagtatgttcaggcatat  
cctccttataccaagttcaccagttcaggtcatcaactggatatacagctgcc  
tgttataactaccagatgccaccgcagtgccctgTGGAGAGCAGAGGA  
GTTATGTT
```

```
>uc008cyu.3_Dazl:474+681 208bp TCCACCACAGTTCAGAGTG AACATAACTCCTCTGCTCTCCA  
TCCACCACAGTTCAGAGTGtttgagtagtccaaatgctgagacttaca  
tcagcctccaaccatgatgaatcctatcaactcagtatgttcaggcatat  
cctccttataccaagttcaccagttcaggtcatcaactggatatacagctgcc  
tgttataactaccagatgccaccgcagtgccctgTGGAGAGCAGAGGA  
GTTATGTT
```

The results obtained using *Dazl*- $\Delta 8$ primers for the real-time RT-PCR:

UCSC In-Silico PCR

The sequences and coordinates shown below are from UCSC Genes, not from the genome assembly.

```
>uc033hel.1_Dazl:616+794 179bp TCCACCACAGTTCAGAGTG CAGTTGTATAAGCCTGGTAG  
TCCACCACAGTTCAGAGTGtttgagtagtccaaatgctgagacttaca  
tcagcctccaaccatgatgaatcctatcaactcagtatgttcaggcatat  
cctccttataccaagttcaccagttcaggtcatcaactggatatacagctgcc  
tgttataaCTACCAGGCTTATACAACCTG
```

The results obtained using *Dazl* RT-PCR primers:

UCSC In-Silico PCR

The sequences and coordinates shown below are from UCSC Genes, not from the genome assembly.

```
>uc008cyv.3_Dazl:715+882 168bp TCCTCCTTATCCAAGTTCACCA TCAGCTCCTGGATCAACTTCAC  
TCCTCCTTATCCAAGTTCACCAgttcaggtcatcaactggatatacagctgc  
ctgtttataactaccagatgccaccgcagtgccctgctggagagcagagg  
agttatggtatacctccggcttatacaactgttaactaccactgcaGTGA  
AGTTGATCCAGGAGCTGA
```

```
>uc033hel.1_Dazl:715+831 117bp TCCTCCTTATCCAAGTTCACCA TCAGCTCCTGGATCAACTTCAC  
TCCTCCTTATCCAAGTTCACCAgttcaggtcatcaactggatatacagctgc  
ctgtttataactaccagcttatacaactgttaactaccactgcaGTGAA  
GTTGATCCAGGAGCTGA
```

```
>uc008cyu.3_Dazl:573+740 168bp TCCTCCTTATCCAAGTTCACCA TCAGCTCCTGGATCAACTTCAC  
TCCTCCTTATCCAAGTTCACCAgttcaggtcatcaactggatatacagctgc  
ctgtttataactaccagatgccaccgcagtgccctgctggagagcagagg  
agttatggtatacctccggcttatacaactgttaactaccactgcaGTGA  
AGTTGATCCAGGAGCTGA
```

Comments 2: In particular, I wanted to know about the DAZL exon was a multiple of 3. Why does the lower band on the western in Figure 7D not change expression to equal the original upper band in the WT?

Response to #2: This is a very good point. Currently, we did not know the exact reason why

the level of DAZL- Δ 8 was not up-regulated to equal the original level of DAZL-FL in the KO testis. However, we might make some assumptions according to current knowledge on RNA splicing and translation. Transcripts with retained introns are often removed by nuclear retention and exosome degradation or nonsense-mediated decay (NMD) to prevent intron-retaining transcripts from being translated into potentially harmful proteins^{1, 2, 3}. In our experiments, apart from the exon 8 skipping, we also detected the retention of intron 7 in *Dazl* in the KO testis. Consistently, we observed the expression of *Dazl* mRNA was decreased in the KO testis. In addition, the translation of *Dazl*-FL and *Dazl*- Δ 8 might not be the same efficiency in testis. Of course, we also could not exclude the possibility that the expression of *Dazl* mRNA might be regulated by *Bcas2* as well.

Comments 3: I notice the exon upstream of DAZL exon 8 is an alternative polyA site -is this used at all?

Response to #3: Using the UCSC Genome Browser and the corresponding sequence provided (a, b), we designed the RT-PCR primers that reside in the exon 2 (shared with other isoforms) and 3' UTR (specific area) (a), respectively. A specific band was successfully amplified from the cDNA of P9 testis with this pair of primer (c). DNA sequencing further confirmed the correction of the sequence (d), suggesting that the alternative polyA site might be used in normal testis. However, 3'-UTR of this isoform is in the intron 7 that was retained and unregulated in BCAS2 null testis (Fig. 6a), leading to be difficult to distinguish these two situation. Thus, we did not describe this isoform in current study.

The result is as follows:

a

b

Full sequence of mouse gene *Dazl* (uc008cyw.2) from UCSC Genome Browser

>uc008cyw.2 (*Dazl*) length=1363

```

ggggcgcgcagccaccgctcagtgactcggcgaccctcagcagcgtcctcgggactccgctgtctccgtggccgtggcctctcttccac
caccgctcggctttttgcccgttggccggctagcgagccaccctcagctagctgccccgtcgtcgttctctccaccctcgaggttttacca
ccgaaactcggcccatctgctgccacaactctgaggtccaaaattcagctgtccaggaggccagcactcagttctcatcagcaac
cacaagtcaggatattgttccagaaggcaaaatcagccaaacacccgtttttgttgaggaaattgatgttaggtgatgaaaccgaaatc
aggagtcttccagatagctcagtaaaagaagtgagataatcactgatcgaactgggtgtgtcgaaggcctatggattgtctattataa
tgacgtggatgtgcagaagatagtagaatcacagataaattccatggtaaaaagctgaaactggccctgcaatcaggaaacaaaattat
gtactatcatgtgcagccagctccttgatttttaactcctcctccaccacagttccagagtggttgagtagtccaaatgctgagacttaacgca
gctccaaacctgatgaactcactcagtagtgcaggcatctccttatacagttcaccagttcagggtcatcactggatcagctgctgctg
ttataactaccaggtattttctttattgaaattttttctcacacatcctgattatggttccccctctgtagtctccacccccactccatccattc
tcacctctataaacaggtttctaagggtcttagtctctattgctgtgaaagaaacaaatggccacagcaattctataaaagaaagacactaa
ttggcctcttaccactactatgaaggataaaatcaaaacaaaactaatgcatcagaataaaacaaaacaaatggaaggaaaaaa
gccaagagaaggtaaaaattagagattcattcaaaactataaattcataaaaatacaaatggaagccataatgtcgcctatgca
tagaacctagtgatgtttgtcaggccccgtgatgtgcaccactcctgtgaattcactgtgagctctgatcattgtttagaaggactgtttctt
gggtctccatcccccaattctcacacactctgcttcttccccaggggtccctgagccctaaaggagggttttaattacagataacta
gggg

```

d

We sequenced the PCR products for verification. The results are shown below.

Sequence primer: GAAGGGCTATGGATTTGTCTC

```

1: TCTGGGGAGTGCAGAGAAGTAGAATCACAGATAAATTTCCATGGTAAAAA
GCTGAAACTGGGCCCTGCAATCAGGAAACAAAATTTATGTACTTATCATG
TGCAGCCACGTCCTTTGATTTTTAATCCTCCTCCTCCACCACAGTTCAG
AGTGTTTGGAGTAGTCCAAATGCTGAGACTTACATGCAGCCTCCAACCAT
GATGAATCCTATCACTCAGTATGTTCCAGGCATATCCTCCTTATCCAAGTTC
ACCAGTTCAGGTATCACTGGATATCAGCTGCCTGTTTATAACTACCAGG
TATTTTTCTTTTATTGAAATTTTTTCTCACACATCCTGATTATGTTTCC
CCTTCTGTAGTCTCCCAACCCCACTTCCATCCCATTCTCACCTCCTT
ATAAACAGGTTTCTAAGGTGTCTTAGTGTCTATTGCTGTGAAGAAACACA
ATGGCCACAGCAATCTTATAAAGAAAGCACTTAATTGGCCTTGCTTACA
CTACTATGTAAGGGATAATAAATCAAAACAAAACAACTAATGCATCAGAATT
AAACAAAACAAATGGAAGGAAAAAGCCCAAGAGAAGGTACAAAAATTAG
AGATTCATTACAAAACCTATAAATCTATAAAAACTAAATGGAAGCCAT
AATGTCTGCCTATGCATAGAACCTAGTGCATGTTTGTGAGGCCCCGTGT
ATGCTGCATCCATCCCTGTGAATTCATGTGAGCTCTGATCATGTTGATTTA
---GGACTGTTTTTCTGGTGTCTCCATCCCTCCAATTCTACACACTATCT
GCTTCTCTTCTCCAGGAA

```

```

2: GGGAGGGGAGAGATAGTAGAATCACAGATAAATTTCCATGGTAAAAAGCTG
AAACTGGGCCCTGCAATCAGGAAACAAAATTTATGTACTTATCATGTGCAGC
CACGTCCTTTGATTTTTAATCCTCCTCCTCCACCACAGTTCAGAGTGTGG
AGTAGTCCAAATGCTGAGACTTACATGCAGCCTCCAACCATGATGAATCCTA
TCACTCAGTATGTTCCAGGCATATCCTCCTTATCCAAGTTCACCAGTTCAGGTG
ATCACTGGATATCAGCTGCCTGTTTATAACTACCAGGTATTTTTCTTTTATTG
AAATTTTTTCTCACACATCCTGATTATGTTTCCCCTCTTGTAGTCTCC
AACCCCACTTCCATCCCATTCTCACCTCCTTATAACAGGTTTCTAAGGTG
TCTTAGTGTCTATTGCTGTGAAGAAACACAATGGCCACAGCAATCTTATAA
AAGAAGCACTTAATTGGCCTTACTACTACTATGTAAGGGATAATAAAT
CAAAACAAAACAAATGCATCAGAATTAACAAAACAAATGGAAGGAAAAAAG
CCCAAGAGAAGGTACAAAAATTAGAGATTCATTACAAAACCTATAAATCTCAT
AAAAATACTAAATGGAAGCCATAATGTCTGCCTATGCATAGAACCTAGTGCA
TGTTTGTGAGGCCCCGTGTGCTGCATCCATCCCTGTGAATTCATGTGAG
CTCTGATCATGTTGATTTAGAAGGACTGTTTTCTGGTGTCTCCATCCCT
CCAATCTTACACACTATCTGCTTCTCTTCCCAGGAA

```

c

Validation of the use of an alternative polyA site upstream of *Dazl* exon 8. (a) Potential alternative splicing isoforms of *Dazl* provided by the UCSC Genome Browser, including the isoform using an alternative polyA site upstream of exon 8 (*Dazl*-short). (b) The sequence of *Dazl*-short from UCSC Genome Browser. (c) A specific band was obtained from the cDNA of P9 testis with primers that reside in the exon 2 and 3' UTR (a). (d) DNA sequencing result of

the amplified PCR products from two normal mice. The sequences labeled in blue were identical to the expected sequence of *Dazl*-short (labeled in red in (b)).

Comments 4: The read height of the RNAseq tracks should be shown.

Response to #4: As you suggested, we have added the read height of the RNA-seq tracks in the revised manuscript (Revised Fig.6a).

Comments 5: How many replicates were used for the RNAseq (Figure 7a) and RT-PCR (Figure 7b). At least the endpoint PCRs should be done in triplicate, and this data should be presented.

Response to #5: The RNA-seq experiment (revised Figure 6a) was performed only one biological sample that was from three control and KO mice. However, RT-PCR (revised Fig. 6b) were used to verify the RNA-seq results that was performed in three independent experiments (a, b and c). We selected (a) as the representative result (revised Fig. 6b) and have added this information into the revised manuscript (P. 38, line19).

The alternative splicing pattern changes of representative genes were verified using RT-PCR. Three independent experiments (a, b, c) were shown.

Comments 6: Are there other experiments that would strengthen the paper further? How much would they improve it, and how difficult are they likely to be? To prove the role of BCAS2 it would be possible to deplete BCAS2 from ES cells, and then monitor effects on DAZL; or carry out minigene experiments to see if expression of BCAS2 changed DAZL splicing patterns. These would both strengthen the connection between BCAS2 and the identified splicing targets. Having said that, if these experiments did not work they would not disprove the main claims.

Response to #6: These are excellent suggestions. Actually, we have already crossed *Bcas2^{Floxed/-}* (*Bcas2^{F/-}*) mice with tamoxifen-inducible cre transgenic (*creER^{T2}*) mice and established the *Bcas2^{F/-}; creER^{T2}* ES cell lines. We treated the ES cells with 4-OH Tamoxifen for 24, 48, 72 hrs, and detected the impact of *Bcas2* depletion on the expression of the two isoforms of *Dazl*. We found that *Bcas2* was dramatically decreased in the ES cells after 24 hrs Tamoxifen treatment (a), however, both *Dazl*-FL and *Dazl*-8 were all up-regulated in the inducible ES cells after 72 hrs tamoxifen treatment (b, c), which was not consistent with that observed in KO testes. These data suggest that the function of *Bcas2* in ES cells might be different from that in testes. On the other hand, we also constructed the *Dazl* minigene which harbored exon 7-intron 7-exon 8-intron 8-exon 9 sequences (2 kb) (d) and transfected it into 293T cells. RT-PCR result showed that exon 8 skipping of *Dazl* pre-mRNA was the mainly form in this system (e), which was in contrast to what was observed in testes, preventing us from probing into the function of *Bcas2* in the splicing of *Dazl* with minigene experiment. We next investigate whether increased *Bcas2* expression can result in opposite effect on *Dazl* splicing. However, no up-regulation of *Dazl*-FL or down-regulation of *Dazl*- Δ 8 was observed when *Bcas2* and *Dazl* minigene were co-transfected into 293T cells (f, g and h). All these data suggest that neither ES cell lines nor minigene experiment in 293T cells might be good models to recapitulate the function of *Bcas2* on *Dazl* splicing in testes. Thus, we did not describe these experiments in current study.

Attempts to confirm the regulatory roles of *Bcas2* on *Dazl* in ES cells and using minigene experiments. (a) Expression of *Bcas2* in *Bcas2*^{F/-}; *creER*^{T2} ES cells treated with 4-OH Tamoxifen for 24, 48, 72 hrs. *Bcas2* was drastically decreased after 24 hrs treatment. (b) Expression of *Dazl-FL* (b) and *Dazl-Δ8* (c) in *Bcas2*^{F/-}; *creER*^{T2} ES cells treated with 4-OH Tamoxifen for 24, 48, 72 hrs. (d) Schematic of the minigene construct. The construct contains exon 7-intron 7-exon 8-intron 8-exon 9 sequences (2 kb). (e) Expression of *Dazl-FL* and *Dazl-Δ8* in 293T cells transfected with various amounts of *Dazl-789* minigene. Note the dominant expression of *Dazl-Δ8* despite the amount of minigene. (f, h and g) Expression of *Bcas2* (f), *Dazl-FL* (h) and *Dazl-Δ8* (g) in 293T cells transfected with 100ng *Dazl-789* minigene and different amount of *Bcas2*.

Comments 7: Are the claims appropriately discussed in the context of previous literature?

Yes

Response to #7: Thank you for your positive comments.

Comments 8: The paper is well written, the abstract/text appropriate. There are some spelling mistakes, e.g. alternative is spelled wrongly in Figure 7.

Response to #8: As suggested, we have corrected the spelling mistakes in the revised manuscript (Revised Fig. 6a). Additionally, we carefully checked the whole manuscript and all modifications were highlighted in the revised version.

Response to Reviewer #2 (Remarks to the Author):

Thanks for the positive comments on our studies. We appreciate your constructive comments to improve our manuscript. The point-by-point responses are as follows:

Comments 1: Data supporting the claim that BCAS2 is highly expressed in spermatogonia is not convincing and the authors should temper their conclusions. Immunostaining is not a quantitative approach. With the level of analysis conducted, statements such as "BCAS2 expression was much stronger in PLZF+ and PLZF- cells" is simply not supported.

Response to #1: We appreciate this comment to improve the presentation of our results with precise statements. According to the suggestion, we have toned down our statements as follows "The BCAS2 expression of BCAS2 in mouse testes" (Revised manuscript P.6, line1). In addition, we have revised the conclusion to that "the expression of BCAS2 was comparatively enriched in PLZF-positive cells" (Revised manuscript P.6, line21-22).

Comments 2: Claims of isolating spermatogonia and somatic cells from P9 testes is not validated. At best, the approach used results in enrichment of the cell types. Thus, the more conservative assessments should be made. Also, the label of SSC in the bar graph of Figure 1b is not accurate. The supposed germ cell population that was isolated is a heterogeneous mix of cells.

Response to #2: We totally agreed with your comments. We have performed experiments to isolate spermatogonia and somatic cells from p9 testes by using OCT4-GFP mouse testes (*Pou5f1*^{tm2(EGFP)^{Jae}}). However, OCT4-GFP did not express highly enough to isolate spermatogonial cells in the mouse line. We also tried to isolate these cells by flow analysis with CDH1 antibody (CST, #3195s, 1:200), however the CDH1 antibody did not work for flow analysis⁴. Because the isolation of spermatogonia does not significantly affect our conclusions in our manuscript, we gave up this attempt and toned down our statements as you suggested.

We have revised the description of “Claims of isolating spermatogonia and somatic cells from P9 testes” into the enrichment of cell types. The enrichment efficiency of cell types was verified using real-time RT-PCR, Western blotting as well as flow cytometry analysis. The information has been added to the revised manuscript (Revised Fig.1b, c, Supplementary Fig.1a, b and manuscript P.6, line11-19). In addition, we have corrected the enriched cells from P9 testes to the fraction of spermatogenic cells (FSPCs) and the fraction of somatic cells (FSCs) (Revised Fig.1b, c, and Supplementary Fig.1a, b).

Comments 3: The RT-PCR methodology for distinguishing between pre-mRNA and mature mRNA for the Tub3 isoforms is not described. Thus, the reader cannot independently assess validity of the approach.

Response to #4: The real-time RT-PCR methodology used to distinguish between and determine the relative amount of pre-mRNA and mature mRNA for the Tub3 is as follows: primers used to determine the amount of *Tuba3a*, *Tuba3b* and *Tubb4b* pre-mRNA were targeted to the 3' end of exon3 and the neighboring 5' end of intron 3. Primers used to

determine the abundance of mature mRNAs of *Tuba3a*, *Tuba3b* and *Tubb4b* were designed to span an exon-exon junction, with *Tuba3a*-mRNA primers annealed to the 3' end of exon4 and 5'end of exon5, *Tuba3b*-mRNA primers annealed to the 3' end of exon1 and 5'end of exon2 and *Tubb4b*-mRNA primers annealed to the 3' end of exon2 and 5'end of exon3. As suggested, we have added this information into the Material and Methods in the revised manuscript (Revised manuscript P.23, line18-21 and P.24, line 1-4).

Comments 5: All assessment of disrupted spermatogenesis is made within the context of the first round of spermatogenesis which may be unique and not reflect the role of BCAS2 or alternative splicing in normal spermatogenesis. In mice, the first meicytes that arise in postnatal life around P8 are known to be derived from a subset of prospermatogonial precursors that do not transit through an undifferentiated spermatogonial state. All other spermatocytes produced in steady-state spermatogenesis are produced from spermatogonial stem cells that arose from a different subset of prospermatogonial precursors. Thus, the first round of spermatogenesis and meicytes are unique compared to subsequent populations. In the current study, the authors only examined the first round spermatocytes and the lack of subsequent second round spermatocytes suggests a defect in spermatogonial differentiation..For these reasons, definitive conclusions about the role of BCAS2 or alternative splicing in meiosis cannot be made. Further experimentation into the cause of impaired spermatogenesis is warranted.

Response to #5: This is a good suggestion. To examined the second round spermatocytes, we analyzed the morphology and histology of testes from two and a half month of control and *Bcas2^{F/-};Vasa-Cre* males. The testes of *Bcas2^{F/-};Vasa-Cre* males were much smaller (Revised Fig.2e, f). Moreover, germ cells were severely reduced and no spermatocytes and spermatids were observed in the seminiferous tubules of *Bcas2^{F/-};Vasa-Cre* males (Revised Fig.2g), indicating impaired spermatogenesis in these males. Immunofluorescence results revealed that only a few MVH-positive cells around the basement membrane survive in the testes of more than two-month old mice (Revised Fig.2h). These data indicate that depletion of *Bcas2* leads to the lack of subsequent second round spermatocytes,

suggesting a defect in spermatogonial differentiation in *Bcas2^{F/-};Vasa-Cre* testes. We have added these information into the revised manuscript (Revised Fig.2e, f, g, h)

Minor Comments

The manuscript contains many typos and grammatical errors. For example, 'alternative' is misspelled multiple time in Figure 7, Western blotting should be capitalized throughout, etc. Incorrect terminology for spermatogenesis is used throughout the manuscript. For example, spermatogonia is often used when the correct term is spermatogonial, and statements indicating that SSCs undergo meiosis and spermiogenesis are not accurate (spermatocytes undergo meiosis and spermatids undergo spermiogenesis).

Response: We have corrected all the mentioned typos and grammatical errors in the revised manuscript. In addition, we have checked the whole manuscript and try our best to polish it.

We have corrected the spelling error in now manuscript (Revised Fig. 6a).

Western blotting have been capitalized in the whole manuscript and highlighted in yellow.

“spermatogonia” has been corrected (Revised manuscript P.3 line9-11 and P.6 line 11-19).

“SSCs undergo meiosis and spermiogenesis are not accurate”. has been corrected with “In mouse testis, spermatogenesis is a complex process involving mitotic cell division, meiosis and spermiogenesis to give rise to haploid spermatozoa” (Revised manuscript P.3, line 9-11).

The revised parts are highlighted in yellow.

Minor Comment: The authors should use the term prospermatogonia in place of gonocyte.

Response: As suggested, we have replaced gonocyte with prospermatogonia (Revised manuscript P.6, line8).

Minor Comment: The title of the manuscript is not accurate. The context is misleading because as written the title indicates the BCAS2 splices spermatogonia, I'm sure the authors mean mRNA splicing here but that is not how the statement is written. In addition, the authors' main conclusion is that BCAS2 and hence alternative splicing influence meiotic progression but this is not reflected in the title.

Response: Thanks for the constructive comment. We have revised the title using “BCAS2 is involved in alternative mRNA splicing in spermatogonia and the transition to meiosis, and male fertility” in the revised manuscript as suggested by reviewer #3 .

Reviewer #3 (Remarks to the Author):

We are pleased with your positive comments and appreciate the helpful and constructive suggestions to improve our manuscript. The point-by-point responses are listed below:

Comment: The first of these is the title - as written the title indicates a fundamental problem in spermatogonial function, while this is technically true, the problems do not manifest until early meiosis. I wonder if 'BCAS2 is (note typo in current draft) involved in alternative mRNA splicing in spermatogonia and the transition to meiosis, and male fertility', would be more informative.

Response: We appreciate this important conceptual comment. The recommended title is indeed more informative and we decide to use this title in the revised manuscript.

Comments: Abstract - that data clearly shows that BCAS2 dysfunction leads to aberrant splicing of many mRNA, several of which could result in sterility. While I agree that mentioning DAZL is informative, the emphasis is distracting. It also begs the question of over

expressing DAZL to correct the defect - this would probably not recover fertility.

Response to #2: We completely agree with your comments. We have deleted the emphasis of DAZL and revised the abstract (Revised manuscript P.2 line 9-14).

Minor points

Minor Comments1: p7 last line. Replace 'stage' with 'type'

Response: “stage” has been replaced with “type” (Revised manuscript P.7 line 19).

Minor Comments 2: The experiments described under the heading 'BCAS2 is critical for germ cell meiosis in mouse testis' needs to be rewritten. At this stage of the manuscript, a second possibility formally exists ie. that spermatogonia are failing to commit to meiosis. Later in the manuscript is possibility is eliminated. Please modify the text at the bottom of p8 to indicate this possibility.

Response: We have reorganized the data and text in the revised manuscript (Revised Fig.3a, b, c, Revised Supplementary Fig.3a, b and Revised manuscript P.8 line 8-22 and P.9 line 1-2). 'BCAS2 is critical for germ cell meiosis in mouse testis' have been deleted in the revised manuscript..

Minor Comments 3: Similarly, the experiments described at the top of p9 are largely an empty analysis. You can see by histology that the cells are missing, so it's no surprise that the markers are all decreased - this could be de-emphasised a little.

Response: As suggested, we have integrated the figure 3 and 4 to support the conclusion that BCAS2 is required for the initiation of meiosis. (Revised Fig.3, Revised Supplementary Fig.3a, b and Revised manuscript P8-P9).

Minor Comments 4: p10 - as these cells have an apoptotic morphology (pyknotic nuclei) it is not possible to definitely tell whether they are have just entered the apoptotic pathway (ie. would be pachytene at d15) or they have been arrested for several days prior to becoming apoptotic. The best you can say is an arrest during early prophase I. Please modify the text. 'Period' rather than 'stage'

Response: We have modified the text according to your suggestion (Revised manuscript P.9 line 14-17). We have corrected the “stage” to “period” (Revised manuscript P.9 line 4 and line 13).

References

1. Jaillon O, *et al.* Translational control of intron splicing in eukaryotes. *Nature* **451**, 359-362 (2008).
2. Gudipati RK, *et al.* Extensive degradation of RNA precursors by the exosome in wild-type cells. *Mol Cell* **48**, 409-421 (2012).
3. Wong JJ, Au AY, Ritchie W, Rasko JE. Intron retention in mRNA: No longer nonsense: Known and putative roles of intron retention in normal and disease biology. *Bioessays* **38**, 41-49 (2016).
4. Ohmura M, Yoshida S, Ide Y, Nagamatsu G, Suda T, Ohbo K. Spatial analysis of germ stem cell development in Oct-4/EGFP transgenic mice. *Arch Histol Cytol* **67**, 285-296 (2004).

REVIEWERS' COMMENTS:

Reviewer #1 (Remarks to the Author):

The authors have answered the questions I raised satisfactorily.

--

Reviewer #2 (Remarks to the Author):

The authors have acceptably addressed my major concerns.

Responds to reviewers:

Reviewer #1 (Remarks to the Author):

The authors have answered the questions I raised satisfactorily.

Responds: We are pleased that you are satisfied our revised manuscript.

Reviewer #2 (Remarks to the Author):

The authors have acceptably addressed my major concerns.

Responds: We appreciate your positive comments for our revised manuscript.